# Colonization of the human gut by bovine bacteria present in Parmesan cheese

Christian Milani [1], Sabrina Duranti[1], Stefania Napoli[2], Giulia Alessandri[3], Leonardo Mancabelli[2], Rosaria Anzalone[2], Giulia Longhi[2], Alice Viappiani[2], Marta Mangifesta[1,2], Gabriele Andrea Lugli[1], Sergio Bernasconi[4], Maria Cristina Ossiprandi[3], Douwe van Sinderen[3,5,6], Marco Ventura[1,4] & Francesca Turroni[1,4]

The abilities of certain microorganisms to be transferred across the food production chain, persist in the final product and, potentially, colonize the human gut are poorly understood. Here, we provide strain-level evidence supporting that dairy cattle-associated bacteria can be transferred to the human gut via consumption of Parmesan cheese. We characterize the microbial communities in samples taken from five different locations across the Parmesan cheese production chain, confirming that the final product contains microorganisms derived from cattle gut, milk, and the nearby environment. In addition, we carry out a human pilot study showing that Bifidobacterium mongoliense strains from cheese can transiently colonize the human gut, a process that can be enhanced by cow milk consumption.

[1] Laboratory of Probiogenomics, Department of Chemistry, Life Sciences and Environmental Sustainability, University of Parma, Parma 43124, Italy. [2] GenProbio srl, Parma 43124, Italy. [3] Department of Veterinary Science, University of Parma, Parma 43126, Italy. [4] Microbiome Research Hub, University of Parma, Parma 43124, Italy. [5] APC Microbiome Ireland, University College Cork, Cork, Ireland. [6] School of Microbiology, University College Cork, Cork, Ireland. These authors contributed equally: Marco Ventura, Francesca Turroni. Correspondence and requests for materials should be addressed to M.V. (email: marco.ventura@unipr.it) or to F.T. (email: francesca.turroni@unipr.it)

Bacteria are ubiquitous microorganisms present in all known environments, including various areas of the mammalian body as well as fermented dairy foods, where they are organized in complex microbial consortia[1]. The microbial ecology of fresh cheese has been extensively investigated[2–4], and has lead to the generally accepted understanding that bacteria residing in milk may contribute beneficially to the organoleptic features of fermented dairy products. In this context, lactic acid bacteria are particularly important due to their positive or negative impacts on fresh and ripened cheese[5–7]. Various studies have focussed on disentangling the microbial communities of the raw dairy material, in particular fresh cow's milk, as well as the composition of the correlated bovine fecal microbiota[7]. Furthermore, it has been shown that the microbial content of cow's milk is greatly influenced by animal husbandry practices, including housing (indoor versus outdoor), feeding and bedding type, which alter the bacterial consortia present on the udder, as well as the milking method, and dust and air in the milking parlor[8–10]. The milk microbiota encompasses a core set of bacteria enriched in members of the Actinobacteria phylum, as well as of the *Strep-tococcus* and *Staphylococcus* genera[7]. Among milk-associated Actinobacteria certain members of the *Bifidobacterium* genus, such as *Bifidobacterium mongoliense*, are commonly present throughout the cheese production chain and in retail cheese products after 21 days of ripening, thus constituting members of the cheese microbiota[11].

Milk has been described as an important vector for vertical transmission of bacteria from mother to corresponding newborn, thereby promoting establishment of the early microbiota in the essentially sterile gut of newborn mammals[12,13]. However, little is known about the potential transfer of bovine-derived microbiota to cheese by means of milk-mediated transmission of a selected consortium of bacteria along the food chain, i.e., from the bovine gut and milk to the final cheese product available to consumers.

In the current study, we applied a microbial ecology-based approach to elucidate the bacterial transmission down to strain level along the production chain of Parmesan cheese, including consumers of the latter product. In this context, we investigated the microbiota composition of a total of 168 samples including stool, litter, and milk samples (MIL) of 50 cows from different husbandries as well as samples of fresh Parmesan cheeses manufactured from the same stocks of milk. Such analyses allowed the identification of a core microbiota that was shown to be horizontally transmitted from dairy cattle to fresh Parmesan cheese. Furthermore, an in vivo pilot study involving 20 Parmesan cheese consumers revealed that members of the cheese microbiota that had been inherited from the production chain colonize and at least temporarily persist in the human gut environment.

## Results

**Metagenomic characterization of the bacterial population**. To investigate the taxonomic composition of the bacterial populations associated with the production chain of Parmesan cheese, we collected a set of samples from five cheese making sites which are highly specialized in the manufacture of Parmesan cheese and located in distinct geographical areas in the provinces of Parma and Reggio Emilia in Italy, named here as P1, P2, P3, RE1, and RE2. Each sample set consisted of 10 dairy cattle faecal samples (CF), 10 cattle litter samples (LIT), 10 MIL, and three samples of Parmesan Cheese at one day of ripening (PC) for a total of 165 samples corresponding to 33 samples per cheese making site (Supplementary Table 1).

Each collected sample was subjected to 16S rRNA gene-based microbial profiling through Illumina sequencing, resulting in a total of 11,622,668 sequenced reads, corresponding to 9,266,623 quality-filtered reads with an average of 56,161 ± 20,455 reads per sample (Supplementary Table 2). The associated microbial profiles obtained at genus level are reported in Supplementary Data 1.

Evaluation of the bacterial biodiversity for each sample was performed by alpha diversity analysis based on the construction of rarefaction curves using 10 data sub-sampling points, as well as on assessment of the Chao1 and Shannon indexes (Fig. 1b, c, Supplementary Figure 1 and Supplementary Figure 2). *T*-tests performed using these data sets showed, as expected, statistically significant (Student $t$ test p-value < 0.05) lower complexity of MIL and PC samples, when compared to CF and LIT samples. This is likely reflecting the typically higher bacterial biodiversity found in the gut of mammals or in environmental samples that are prone to contamination by faecal material[14], when compared to milk and cheese samples[15,16] (Fig. 1b, c, Supplementary Figure 1 and Supplementary Figure 2). Moreover, all obtained rarefaction curves based on 10 sub-sampling of the 165 datasets tend to plateau, indicating that the data sets obtained from Illumina sequencing are enough to sufficiently cover the biodiversity of the samples and to reconstruct a comprehensive taxonomic profile (Supplementary Figure 1 and Supplementary Figure 2).

Taxonomic data of the 165 samples was also used to perform a Principal Coordinate Analysis (PCoA), revealing, as expected, clustering of the datasets based on the sampled matrix (Supplementary Figure 3 and Fig. 1a). Nevertheless, partial overlap was observed for MIL and LIT samples collected from the same production site (Supplementary Figure 4), suggesting contamination of mammary glands with the litter during the housing of the animals. Moreover, individual beta-diversity analysis of CF, LIT, MIL, and PC samples evidenced a limited effect of sampling site, as indicated by absence of clustering in the PCoA representation based on the geographic locations (Supplementary Figure 3) (Supplementary Figure 4). This was verified by performing an Adonis test for each matrix, resulting in a $p$-value > 0.05 for all conditions tested.

The CF samples are dominated by typical gut colonizers of ruminants[17] such as members of the *Ruminococcaceae*, *Rikenel-laceae,* and *Lachnospiraceae* families with average abundances of 44.6%, 11.3% and 9.2%, respectively (Supplementary Data 1), while aerobic bacteria such as *Corynebacteriaceae*, *Staphylococca-ceae,* and *Aerococcaceae*[17–19] represent the most abundant bacteria found in LIT samples with an average abundance of 16.2%, 12.3% and 6.1%, respectively (Supplementary Data 1). Furthermore, all the MIL samples revealed a high average abundance of *Lactobacillaceae*, *Bifidobacteriaceae*, *Corynebacter-iaceae*, and *Staphylococcaceae* corresponding to 22.4%, 13.5%, 7.5%, 6.9% of the total average bacterial population, respectively (Supplementary Data 1). Conversely, PC samples are dominated by genera abundant in whey, which is used as a source of natural starter cultures in Parmesan cheese making, i.e., *Lactobacillaceae* and *Streptococcaceae* with an average relative abundance of 90.3% and 8.7%, respectively (Supplementary Data 1).

While these data confirm previous observations regarding the occurrence of the microbial genera in these ecological niches[14–16], we also explored the notion that bacteria may be transmitted across the Parmesan cheese production chain by the assessment of putative shared bacterial taxa. Notably, evaluation of bacterial genera detected in at least 70% of the samples (prevalence > 70%) of each pool of collected matrices, i.e., CF, LIT, MIL, and PC, revealed that 37 of the 84 genera with a prevalence > 70% profiled in CF datasets are shared with LIT samples, thus indicating extensive persistence of faecal contaminants in litters. Moreover, 15 of the 37 genera shared between CF and LIT datasets were also identified in MIL samples. Notably, these shared bacterial taxa represent 68.5, 17.4, and 16.7% of the total average relative

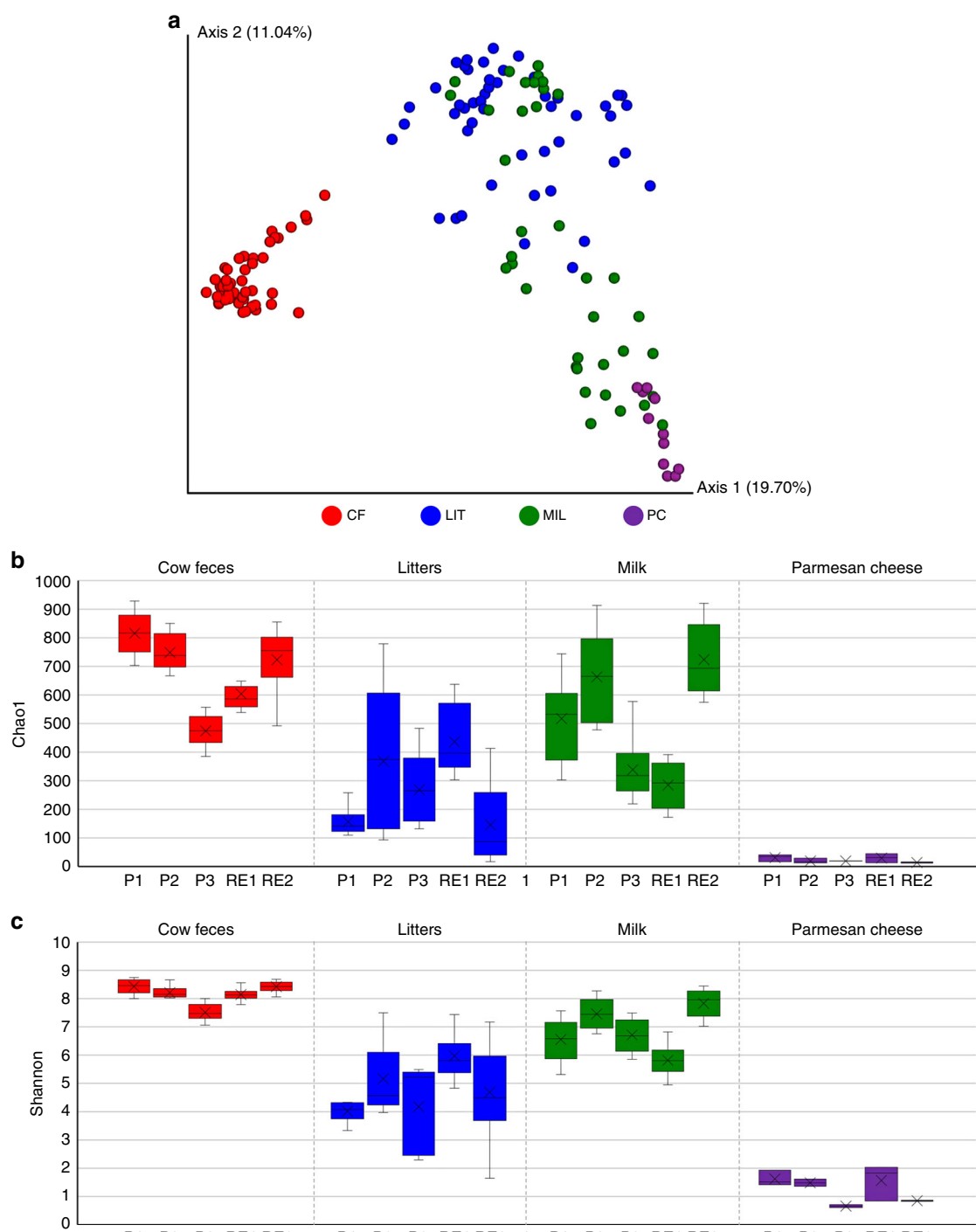

**Fig. 1** 16S rRNA gene-based microbial profiling of the 165 samples included in this study. Panel **a** visualizes the PCoA representation of the beta-diversity observed with samples colored based on matrix type: cow feces (CF), litters (LIT), milk (MIL), Parmesan cheese (PC). Panels **b** and **c** show whisker plots based on Chao1 and Shannon alpha-diversity, respectively, observed for a sub-sampling of 26,666 reads. The boxes represent 50 % of the data set, distributed between the first and third quartiles. The Median separates the boxes into the interquartile range, while the X represents the Mean. The lines extending vertically outside of the boxes show the outlier range. Source data are provided as a Source Data file

abundance of the bacterial population harbored by CF, LIT, and MIL samples. Such intriguing results reflect the proposed role of milk as a natural vector of gut bacteria to promote gut colonization of breastfeeding offspring[12,20]. As expected, data collected in the current study suggest that gut bacteria may reach and colonize the mammary gland not only through a systemic route, as previously hypothesized[21], but also by means of direct contact with fecal-contaminated environments such as litters,

thereby promoting exchange of bacteria between cohabitating cows as reported previously for humans, as well as humans and their companion animals[22].

Evaluation of bacterial taxa that were detected in PC as well as CF, MIL, and LIT samples of, at least, a cheese making site with a prevalence of >70% in at least three matrices resulted in the identification of 13 bacterial genera. Among the latter, two taxa, i.e., *Bifidobacterium* and members of *Lachnospiraceae* family,

were shared across four cheese making sites and an additional three, i.e., *Bacteroides*, *Streptococcus*, and U. m. of *Actinobacteria* class, were shared across three Parmesan cheese production plants. Intriguingly, due to their detection in all sampled matrices, these taxa represent the core microbiota of the Parmesan cheese production chain, including faecal samples of dairy cattle, milk, and fresh cheese. Notably, raw-milk cheeses, such as Parmesan cheese, are produced using non-pasteurized milk[23]. Thus, despite the fact that microbial profiling based on 16S rRNA gene sequences cannot distinguish between DNA of viable or dead cells, it is expected that many of the bacterial taxa constituting the core microbiota of the Parmesan cheese production chain will in fact reach dairy consumers as viable microbial cells due to the absence of thermal treatments during Parmesan cheese production. In this context, it is worth mentioning that the dominance of *Lactobacillus* and *Streptococcus* in the average PC microbiota (Table 1) (Supplementary Data 1) is supported by their addition during whey inclusion, thus increasing the load of *Lactobacillus* and *Streptococcus* strains transmitted from CF, LIT, and MIL samples. Nevertheless, as was demonstrated by shotgun sequencing analyses and by the use of PCR-strain-specific primers (see below), strains of these two genera assembled in PC samples are also found in CF, LIT, and MIL samples. These findings therefore confirm the results obtained from 16S rRNA gene microbial profiling and core genome reconstruction. All these aforementioned bacteria may thus impact on flavor, taste, and texture of both fresh and ripened cheese, while they may also use cheese as a vector for gut colonization.

**Tracing of 16S rRNA gene and bifidobacterial ITS-based OTUs.** Evaluation of the Operational Taxonomic Unit (OTUs) distribution (Supplementary Data 2), when generated at 100% identity and with a prevalence >70% in the pools of collected matrices retrieved from the same cheese production site, revealed, as expected, that CF and LIT represent those environments possessing the highest average number of shared OTUs, i.e., 49, of which an average of 11 were also found in MIL samples (Fig. 2a). These findings support the assumption that litters promote transmission of gut bacteria to mammary glands. Furthermore, despite the low bacterial biodiversity, which is characteristic of cheese and MIL, OTU analysis indicated that an average of seven taxonomic units are shared between these two matrices (Fig. 2a).

Regarding OTUs shared by CF, LIT, MIL, and PC samples of each sampling site, an average of four shared taxonomic units was observed. These OTUs were further analyzed through manual classification to the lowest reliable taxonomic rank based on alignment of their reference sequence to the NCBI nt database (Fig. 2b). Intriguingly, these results showed that the number of bacterial taxa shared across the cheese production chain as detected by 16S rRNA gene microbial profiling may vary considerably, ranging from two to eight, reflecting the high variation encountered in cheese manufacturing sites (Fig. 2b). In this context, it should be realized that the number of reads obtained for each sample may be responsible for putative biases in the number of OTUs detected and that an increased sequencing depth is expected to identify additional taxa shared across the cheese production chain.

Altogether, these results highlight that the functional role of bacterial taxa distributed across the Parmesan cheese production chain is still far from fully understood and requires further exploration through isolation and functional genomic characterization efforts.

To further validate the suspected presence of identical bacterial species across the Parmesan cheese production chain, we focused on the genus *Bifidobacterium* due to the availability of a genus-specific profiling tool based on the Internally Transcribed Spacer (ITS) sequence, allowing a very accurate bifidobacterial identification at Phylotype level, i.e., between species and strain level[24]. Sequencing of all CF, MIL, and PC samples resulted in a total of 4,580,699 quality-filtered reads, with an average of 39,832 reads per sample (Supplementary Table 3).

Interestingly, taxonomic profiles obtained from CF and MIL samples (Supplementary Data 3) confirm previous observations regarding the ecology of bifidobacterial species in faecal and MIL[20]. In this context, detection of *Bifidobacterium longum* subsp. *infantis* in bovine MIL supports the notion that bifidobacterial species have evolved to colonize a wide range of mammals, which may reflect strain-level adaptation to specific hosts, as previously suggested[20]. Moreover, analysis of PC samples showed that *Bifidobacterium mongoliense* represents, on average, 33.6% of the overall bifidobacterial population with a prevalence of 93.3%, while *B. adolescentis*, *B. bifidum*, *B. breve*, *Bifidobacterium crudilactis*, *B. longum* subsp. *longum*, and *B. pseudolongum* subsp. *globosum* were shown to be present at an average prevalence level of 82.7% and represent, altogether, a total of 37.7% of the average bifidobacterial population harbored by fresh Parmesan cheese (Supplementary Data 3).

Furthermore, analysis of OTUs with a prevalence of >70% in CF, MIL, and PC samples collected from the same cheese plant indicated extensive transmission of particular bifidobacterial phylotypes through the cheese production chain (Fig. 3a). Notably, CF and MIL samples were shown to contain an average of 24 identical OTUs, while on average 56 identical OTUs were shown to be present in MIL and PC matrices (Fig. 3a). These data reflect the high relative abundance of bifidobacteria observed in milk through 16S rRNA gene-based profiling (average of 12.6% in MIL samples) and show that bifidobacterial taxa derived from milk consistently participate in defining the bacterial biodiversity harbored by fresh Parmesan cheese despite their low relative abundance compared to the overall bacterial population (average of 0.02% as based on 16S rRNA gene profiling data). Moreover, integration of the latter results revealed that an average of 11 OTUs are shared by CF, MIL, and PC samples (Fig. 3a).

Manual classification of OTUs to the lowest reliable taxonomic rank, based on an updated version of the bifidobacterial ITS database[24] (http://probiogenomics.unipr.it/pbi/), revealed that phylotypes of nine bifidobacterial species are shared by at least two matrices (Fig. 3b). Intriguingly, among the latter taxa, *B. mongoliense*, *B. bifidum*, *B. longum* subsp. *longum*, and *B. pseudolongum* subsp. *globosum* are shared by CF, MIL and PC matrices of all assessed Parmesan cheese production sites, with the exception of *B. bifidum* being below the 70% prevalence cut-off (observed prevalence of 30%) in CF samples of the Reggio Emilia 2 sampling site (Fig. 3b). Notably, *B. mongoliense* has a major relevance in cheese making due to its widespread presence in raw milk and fresh cheese[11]. For this reason, the ecology of this bifidobacterial species was further explored in the current study in order to shed light on its potential transmission to the human gut following cheese consumption.

**Strain tracing across the Parmesan cheese production chain.** Despite the use of 16S rRNA- or ITS-based sequences for metagenomic profiling purposes, which suggests bacterial transmission across the Parmesan cheese production chain, a strain-specific validation is required to confirm the notion of bacterial transmission. Thus, in depth shotgun metagenomics sequencing of a subset of eight samples was performed. The latter encompassed two samples for each matrix analyzed, which are characterized by high relative abundance (>19.8%) of bacterial species whose OTUs were observed to be shared by CF, LIT, MIL, and PC

**Table 1 Average relative abundance of taxa with a prevalence of >70% in at least three matrices**

| | | Alistipes | Bacteroides | Bifido-bacterium | Coryne-bacterium 1 | Lactobacillus | Ruminoc-occaceae UCG-005 | Ruminoc-occaceae UCG-010 | Streptoc-occus | U. m. of Actinobacteria class | U. m. of Clostridiaceae 1 family | U. m. of Lachnospiraceae family | U. m. of Peptostrept-ococcaceae family | U. m. of Ruminoc-occaceae family |
|---|---|---|---|---|---|---|---|---|---|---|---|---|---|---|
| P1 | CF | 4.704% | 3.640% | 0.024% | - | 0.003% | - | - | 0.001% | 0.003% | 0.138% | - | 0.260% | 8.920% |
| P1 | MIL | 0.237% | 0.233% | 0.524% | - | 2.354% | - | - | 0.381% | 2.601% | 0.137% | - | 0.697% | 0.669% |
| P1 | LIT | 0.207% | 0.308% | 27.007% | - | 47.502% | - | - | 4.669% | 0.311% | 0.156% | - | 0.468% | 0.168% |
| P1 | PC | 0.009% | 0.020% | 0.046% | - | 87.706% | - | - | 10.707% | 0.314% | 0.012% | - | 0.023% | 0.013% |
| P2 | CF | - | - | 0.996% | - | - | 15.502% | 8.951% | 0.065% | 0.008% | - | 10.097% | - | - |
| P2 | MIL | - | - | 0.617% | - | - | 1.576% | 0.266% | 0.099% | 4.563% | - | 1.918% | - | - |
| P2 | LIT | - | - | 26.584% | - | - | 0.990% | 0.358% | 5.349% | 1.286% | - | 2.523% | - | - |
| P2 | PC | - | - | 0.038% | - | - | 0.007% | 0.002% | 12.915% | 0.266% | - | 0.005% | - | - |
| P3 | CF | - | 5.786% | 0.120% | 0.002% | - | - | - | - | - | - | 9.505% | - | - |
| P3 | MIL | - | 0.180% | 0.023% | 11.698% | - | - | - | - | - | - | 0.258% | - | - |
| P3 | LIT | - | 0.750% | 3.927% | 10.005% | - | - | - | - | - | - | 2.902% | - | - |
| P3 | PC | - | 0.111% | 0.012% | 0.043% | - | - | - | - | - | - | 0.016% | - | - |
| RE1 | CF | - | 5.089% | - | - | 0.015% | - | - | - | 0.001% | 0.243% | 11.009% | 0.221% | - |
| RE1 | MIL | - | 0.459% | - | - | 0.663% | - | - | - | 7.883% | 0.220% | 2.242% | 1.001% | - |
| RE1 | LIT | - | 4.959% | - | - | 21.203% | - | - | - | 1.647% | 1.054% | 5.815% | 5.182% | - |
| RE1 | PC | - | 0.063% | - | - | 85.654% | - | - | - | 0.095% | 0.028% | 0.053% | 0.014% | - |
| RE2 | CF | - | - | 0.013% | - | - | - | - | 0.007% | - | - | 3.409% | - | - |
| RE2 | MIL | - | - | 0.122% | - | - | - | - | 0.043% | - | - | 1.053% | - | - |
| RE2 | LIT | - | - | 2.600% | - | - | - | - | 1.166% | - | - | 1.394% | - | - |
| RE2 | PC | - | - | 0.016% | - | - | - | - | 6.869% | - | - | - | 0.033% | - |

samples collected from the same cheese manufacturing site (listed in Fig. 2b). Shotgun metagenomics sequencing produced a total of 154,053,240 reads, ranging from 12,273,062 to 37,612,529 reads for individual samples (Supplementary Table 4). These reads were then processed to remove bovine DNA through read mapping to the complete *Bos taurus* genomic sequence. The resulting filtered reads were then used for assembly and species-specific genome reconstruction using the bioinformatic platform METAnnotatorX[25]. In order to ensure high accuracy in taxonomic classification of the contigs, only assembled sequences with a length of >3000 bp were retained. Due to the high abundance of eukaryotic DNA contained in MIL, their assembly provided highly fragmented genomes that were not further analyzed. In contrast, the bacterial species with the most complete genomic sequence assembled from the sequences obtained from each PC, LIT, and CF sample were used for the design of strain-specific primers (Supplementary Table 5) (Fig. 4). In detail, among the assembled sequences sharing the same taxonomy at species level, the largest contig, representing a chromosomal region of a specific strain, was employed for such primer design. These primers were tested for specificity by comparative sequence analysis to any known sequence contained in the NCBI nt database employing the Primer BLAST web tool[26]. This analysis did not return any hits for the tested primers.

The strain-specific primers were used for a PCR reaction in combination with bacterial DNA extracted from samples collected from the corresponding cheese production site (Fig. 4). The use of strain-specific primers based on bacterial sequences that were retrieved by metagenomic sequencing of CF, LIT, MIL, and PC samples of a given production site allowed us to precisely trace the occurrence of certain bacterial strains across the production chain corresponding to such a cheese manufacturing facility. Intriguingly, PCR results revealed that 14 species were successfully traced in at least two matrices across the Parmesan cheese production chain of the cheese making plant from where they had been identified (Fig. 4). Amongst these species, *Corynebacterium stationis*, *Prevotella ruminicola,* and *B. mongoliense* were detected in all sampled matrices (CF, LIT, MIL, and PC), while seven species were detected in PC and at least one additional matrix (Fig. 4). Although the strain-specific primer tracing approach offers high sensitivity, it may nonetheless not provide high enough specificity in case the amplified DNA sequence represents a conserved region between closely related strains. For this reason, and in order to substantiate our observations at strain level, we used reads that mapped with a 100% read identity and read coverage cut-off to evaluate with single nucleotide polymorphism (SNP) accuracy the distribution of genomes assembled from samples collected in the P1 and P2 cheese making sites (Fig. 4). Notably, mapping of reads representing the available PC dataset collected in P1 onto the assembled (from the P1 data sets) genomes of *B. mongoliense*, *C. stationis*, *Corynebacterium variable*, and *Pseudoclavibacter soli* resulted in fully mapped reads in all cases (Supplementary Table 6). Similarly, read mapping of PC dataset of P2 onto bacterial chromosomes assembled from P2 datasets, and belonging to *Lactobacillus delbrueckii, Lactobacillus helveticus, Atopostipes suicloacalis, Oligella ureolytica, Jeotgallicoccus psychrophilus,* and *Streptococcus thermophilus* species, produced confirmative results in all cases (Supplementary Table 6). In this context, the metagenomic read mapping approach provides higher specificity compared to the strain-specific PCR approach of metagenomic DNA, yet at the cost of lower sensitivity, as evidenced by the low coverage obtained for taxa present at low relative abundance in the analyzed bacterial population (Supplementary Table 6), i.e., *Corynebacterium variable, Atopostipes suicloacalis, Jeotgallicoccus psychrophilus*, and *Oligella ureolytica*.

In order to support the strain-level tracing across the Parmesan cheese production chain, we exploited a SNP profiling analysis using the genomes assembled from samples collected in the P1 and P2 cheese manufacturing sites (Fig. 4) as reference sequences (Supplementary Data 4). Such in silico analyses were carried out using BWA aln software and the metaSNV bioinformatic tool. The obtained SNP profiles demonstrated that the assembled contigs fully correspond to the most abundant strain as indicated by homogeneous profiles and absence of high frequency SNPs when reads corresponding to datasets used for assembly were mapped (Supplementary Data 4). Moreover, mapping of the additional dataset collected from the same cheese production site generated a very similar SNP profile for all 10 species. These findings confirm that the different sampled matrices, i.e., LIT and PC, of the same cheese making site share the same bacterial strain (Supplementary Data 4). Remarkably, mapping of datasets obtained from samples collected in the cheese production site where a different strain was expected indeed resulted in markedly different SNP profiles (Supplementary Data 4), which confirmed the PCR results achieved by strain-specific primers, i.e., the absence of any strain-specific amplicon. These SNP profile observations were also supported by SNP distance analysis (Supplementary Data 4). To obtain further, in vitro validation of strain transmission events across the Parmesan cheese production chain, bacterial isolation, and targeted microbial genomic sequencing attempts were carried out (see below).

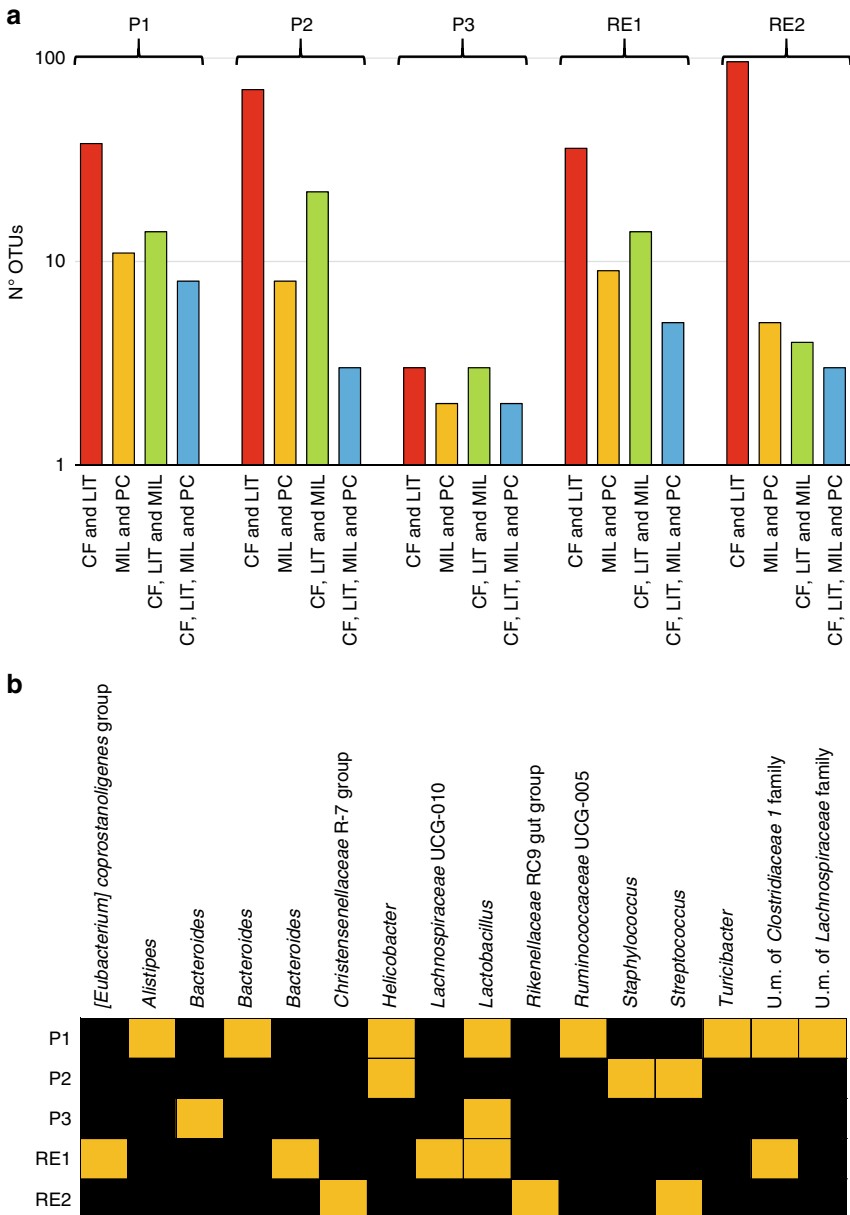

**Fig. 2** 16S rRNA gene-based OTU analysis surveying possible microbial transmission events. Panel **a** shows a bar plot of the number of 16S rRNA gene OTUs shared between multiple matrices (prevalence of >70%) collected from the same cheese making site. Panel **b** reports a heat map of the bacterial taxa for which OTUs have been found to be shared with a prevalence of >70% by CF, LIT, MIL, and PC samples of the same cheese production site. Source data are provided as a Source Data file

The partial metagenome-based genomic assemblies used for primer design were subjected to prediction and analysis of their glycobiomes (Fig. 4). Data collected revealed that strains found in multiple matrices encompass genes involved in the metabolism of milk-related compounds that may be responsible for their ability to survive across the cheese production chain.

Intriguingly, these data represent the first strain-level validation of horizontal transmission of bacteria across the Parmesan cheese production chain and reveals that (fresh) cheese may act as a transmission vehicle of such bacteria to humans. Notably, this may have a major impact on the genomic diversity in the developing microbiota during early life[27]. Moreover, the presence of dairy cattle's gut and milk bacteria in cheese, along with environmental bacteria found in litters may be pivotal to understand the peculiar organoleptic features that distinguish

the cheese made by different producers, since such bacteria may be expected to play a role in cheese ripening and maturation.

**Isolation of horizontally transmitted strains.** Due to the relevance of lactobacilli in the dairy industry[28], this genus was chosen as the target of our isolation and genomic sequencing efforts that included all collected matrices of cheese production sites P1 and P2. In detail, this culture-dependent approach was based on *Lactobacillus*-specific growth media/conditions, which allowed the isolation of 64 lactobacilli, and culminating in the identification of two strains, i.e., *L. delbrueckii* LDELB18P1 and *L. delbrueckii* LDELB18P2, that are shared by CF, MIL, LIT, and PC samples of the P1 and P2 cheese manufacturing sites, respectively. For each species, the isolates sampled from the CF, MIL, LIT, and

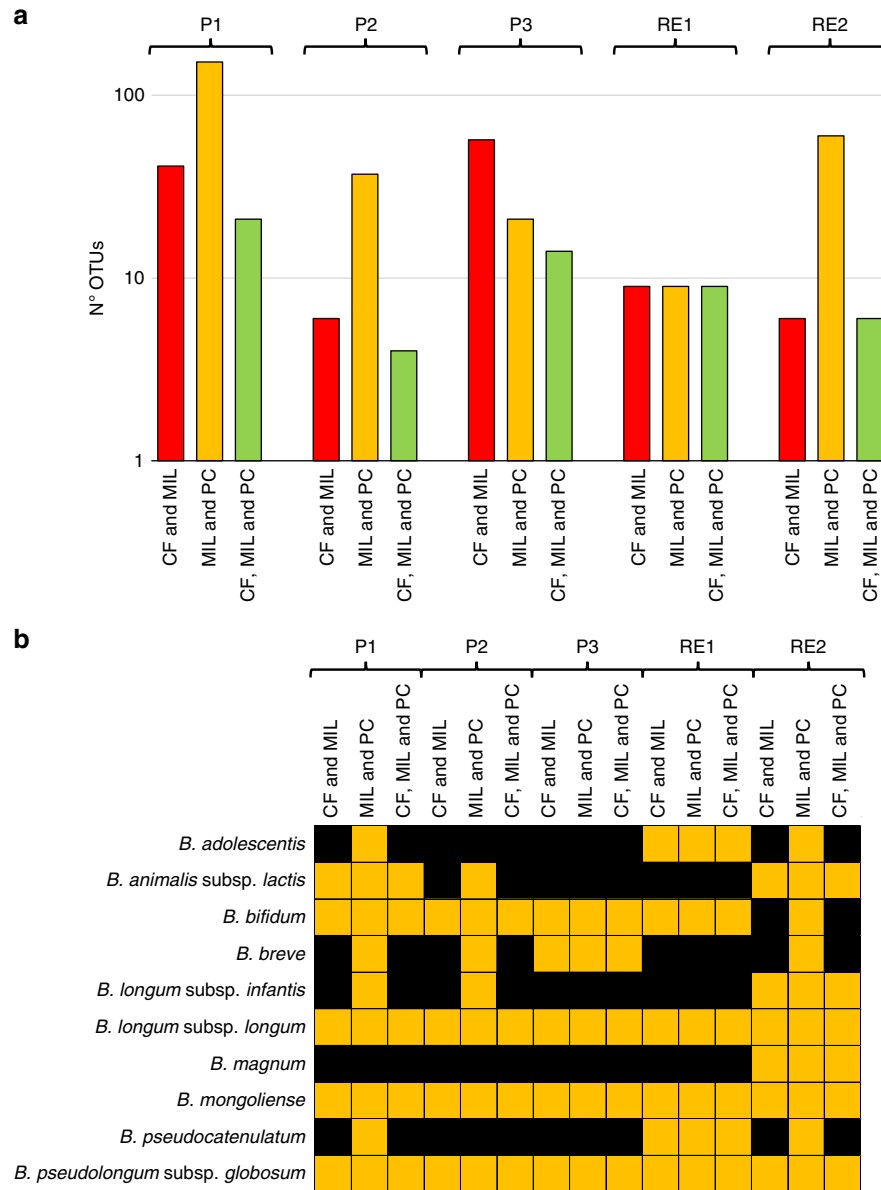

**Fig. 3** Bifidobacterial ITS-based profiling of CF, MIL, and PC samples. Panel **a** shows a bar plot displaying the number of OTUs shared between multiple samples collected from the same cheese making site. Panel **b** illustrates by means of a heat map the bifidobacterial taxa for which OTUs were found to be shared with a prevalence of >70% between CF, MIL, and PC samples of the same cheese making site. Source data are provided as a Source Data file

PC matrices were screened with strain-specific primers designed on contigs assembled from shotgun metagenomics data (Supplementary Table 5) (Fig. 4). For each matrix, i.e., CF, MIL, LIT, and PC, an isolate showing a positive PCR product was subjected to shotgun sequencing. Alignment of all the reconstructed genomes isolated from the same cheese making site revealed an average nucleotide identity (ANI) value of 100%, thus confirming that they represent the same strain. Moreover, genomic alignment of *L. delbrueckii* LDELB18P1 and *L. delbrueckii* LDELB18P2 provided an ANI value of 98.89%, thereby validating that they represent distinct strains, and alignment of the contig exploited for primer design to the corresponding isolate's genome showed a nucleotide identity of 100%, as expected.

Furthermore, due to the relevance of bifidobacteria as health-promoting bacteria and the high prevalence of *B. mongoliense* species in raw milk and fresh cheese[11], the same culture-dependent workflow was exploited for the isolation of a strain shared by CF, MIL, LIT, and PC matrices collected from cheese production site P1, starting from 18 bifidobacterial isolates obtained through growth on selective media. DNA extracted from all these bifidobacteria isolates was subjected to PCR amplification using the strain-specific primers (previously designed based on assembled shotgun metagenomics data; Supplementary Table 5). These analyses allowed the identification of a *B. mongoliense* strain, which, based on its genome sequence (see below), is shared by the CF, MIL, LIT, and PC matrices collected from cheese production site P1. A *B. mongoliense* isolate collected from each of the CF, MIL, LIT, and PC matrices of P1, which generated an expected amplification product with the above-mentioned strain-specific primers, was subjected to genome sequencing. Notably, comparative genomics analyses revealed that all these *B. mongoliense* isolates were isogenic, i.e., they displayed an ANI value of 100%. This shared bifidobacterial strain was named *B. mongoliense*

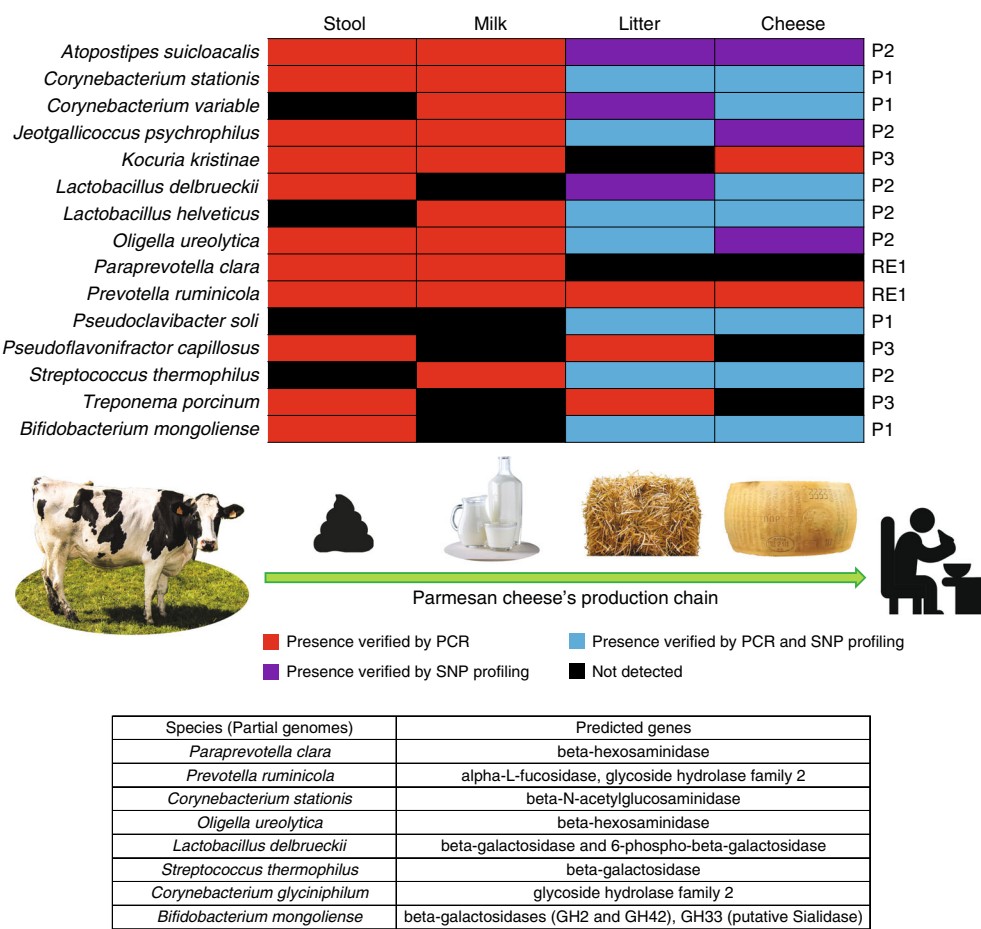

**Fig. 4** Strain-specific tracing of bacterial transmission events. The heat map reports in red, purple, and blue the successful strain-specific identification of genomes assembled from shotgun metagenomics data by means of PCR, SNP profiling or PCR and SNP profiling, respectively. Presence of these strains was tested in the CF, LIT, MIL, and PC matrices sampled from the same cheese production site where they were identified. The production site is reported on the right side of the heat map. The table reports genes involved in utilization of milk-related carbohydrates predicted in the partial genomes of the traced strains. Source data are provided as a Source Data file

BMONG18 and further genomically and functionally characterized.

**Transcriptome analysis of *B. mongoliense* BMONG18 strain.** Due to the relevance of bifidobacteria in the food industry and the prevalence of *B. mongoliense* in raw milk and fresh cheese[11], we decided to gain insights into the genetic and transcriptional adaptation of BMONG18 to growth in milk and to exploit this strain for a pilot study aimed at exploring its transmission and persistence in humans (see below).

Interestingly, analysis of the genetic potential of the isolated BMONG18 strain revealed the presence of an extensive arsenal of genes involved in the metabolism of milk-related carbohydrates (Fig. 4).

Transcriptome analysis was performed by means of an RNAseq experiment involving three biological replicates of *B. mongoliense* BMONG18 grown on MRS and three biological replicates of *B. mongoliense* BMONG18 cultivated on bovine milk (Supplementary Table 7). Intriguingly, transcriptome data revealed that growth on cow milk leads to a more than three-fold, based on RPKM evaluation, increased transcription of 41 genes encoding transporters that were predicted to be involved in cellular internalization of glycans and 17 genes encoding enzymes involved in the metabolism of carbohydrates, including beta-galactosidase and exo-alpha-sialidase genes that participate in

bovine milk oligosaccharide degradation[20] (Supplementary Table 7). Intriguingly, we also observed that growth in bovine milk caused increased transcription of the tight adherence (TAD) locus, i.e., six genes responsible for Type IV pilus assembly, which have previously been shown to aid in host colonization of the human gut commensal *Bifidobacterium breve* UCC2003 in the mammalian intestine[29,30], while also being implicated in host colonic cell proliferation[30]. Altogether, these results suggest a genetic adaptation of *B. mongoliense* BMONG18 to bovine milk and that this nutritious secretion activates specific transcriptional patterns aimed at supporting gut colonization of and host communication by this species.

Notably, *B. mongoliense* has been reported as a member of the microbiota of many ripened cheeses[11], thus representing an obvious candidate to test the hypothesis of human colonization vectored by bovine milk-based cheese products[31,32].

**Transmission of bacteria from Parmesan cheese to the human gut.** To evaluate if bacteria transmitted across the Parmesan cheese production chain can colonize and persist in the human gut, an in vivo pilot study involving a total of 20 healthy individuals was performed. The enrolled subjects were requested to eat the recommended daily dose of fresh Parmesan cheese[33], i.e., 45 g/day, for seven days. Notably, qPCR with species-specific primers (Supplementary Table 5) did not reveal occurrence of *B.*

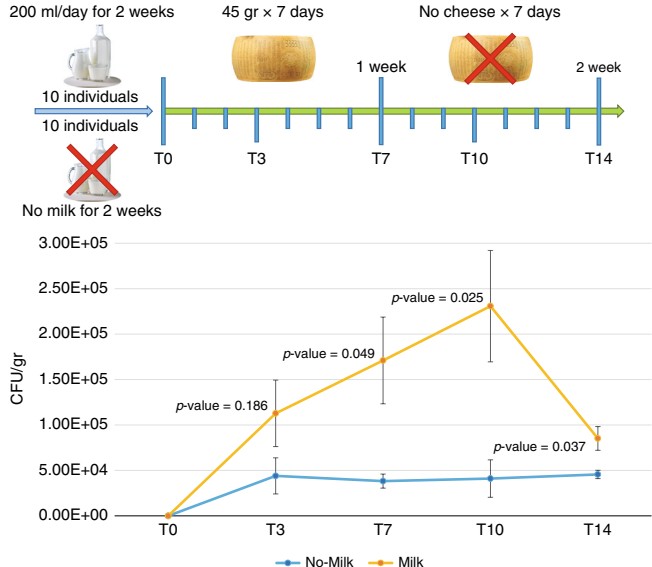

**Fig. 5** Evaluation of the performance of *B. mongoliense* BMONG18 in achieving human gut colonization and persistence. The graph shows the average abundance of *B. mongoliense* BMONG18 observed in the enrolled 10 Milk and 10 No-Milk individuals at multiple time points, as described in the study plan schematized in the upper part of the image. Error bars represent the standard deviation. P-values obtained from Student *t* test between Milk and No-Milk individuals are reported in the graph for each sampling point. Source data are provided as a Source Data file

*mongoliense* at the start of the pilot study. Each enrolled individual was provided with seven 45 g portions obtained from the same cheese produced in plant P1 and the presence of *B. mongoliense* BMONG18 in the portioned fresh Parmesan cheese was verified using a strain-specific primer pair (Supplementary Table 5), indicating an abundance of 3.66E+05 CFU/g. These primers were also used to evaluate the presence and absolute abundance of *B. mongoliense* BMONG18, by means of qPCR, in the stool samples of the 20 enrolled individuals during fresh Parmesan cheese consumption and for seven subsequent days (Fig. 5). In order to evaluate the role of milk consumption in supporting colonization and persistence of *B. mongoliense* BMONG18, 10 individuals drank 200 ml of pasteurized cow's milk each day during and after the seven days of fresh Parmesan cheese consumption (Milk group), while 10 individuals didn't consume milk or any other dairy product (with exception of Parmesan cheese) during any phase of the study (No-Milk group) (Fig. 5).

The evaluation of the modulatory effects of Parmesan cheese consumption on the gut microbiota was explored through 16S rRNA gene microbial profiling of T0, T7, and T14 samples collected from all the enrolled Milk and No-Milk individuals (Supplementary Table 8) (Supplementary Data 5). Rarefaction curves showed extensive biodiversity coverage and no significant differences were identified between time points and groups by means of paired *T*-test (Supplementary Data 5). Statistical analysis through paired *T*-test of 16S rRNA gene microbial profiling data obtained for all faecal samples collected at T0, T7, and T14 revealed the absence, as expected, of main differences in the gut microbial profiles of either No-Milk or Milk groups (Supplementary Figure 5). In this context, minor changes at the genus-level of the gut microbiota composition due to the daily intake of 45 g of Parmesan cheese were probably masked by the individual temporal variation in gut microbiota composition, as previously reported[34]. Moreover, taxa that may have been

modulated by Parmesan cheese consumption may fall below the limit of detection of 16S rRNA gene microbial profiling approach. In this context, it is worth to mention that gut bacterial taxa may exert major functional roles with relevant impact on host's physiology disregarding their low relative abundance[35]. For this reason, *B. mongoliense* BMONG18-specific primers were employed to precisely trace the colonization by this particular strain at T0, T3, T7, T10, and T14 (Fig. 5).

Notably, *B. mongoliense* BMONG18 was detected in the feces of all enrolled individuals during the seven days of cheese consumption, as expected, with an average abundance of 1.71E+05 CFU/g in the Milk group and 3.82E+04 CFU/g in the No-Milk group at T7 (Fig. 5). Intriguingly, individuals of the Milk group, on average, showed higher abundance, i.e., CFU per g of feces, of *B. mongoliense* BMONG18 as compared to the No-Milk group with Student *t* test *p*-values of 0.049, 0.025 and 0.037 at T7, T10 and T14, respectively (Fig. 5). This indicates that daily bovine milk consumption supports efficient colonization of *B. mongoliense* in the human gut. Nevertheless, tracing of *B. mongoliense* BMONG18 in individuals of the Milk group revealed a decrease of *B. mongoliense* abundance at T14, i.e., 1 week after removal of Parmesan cheese from the daily diet (Fig. 5), suggesting that weekly ingestion of *B. mongoliense* through Parmesan cheese consumption is probably needed to maintain its long-term persistence in the human gut.

Notably, *B. mongoliense* was isolated from feces collected from three individuals at the end of the second week (sample T14). The genomes of these three isolates were sequenced and shown to be identical to the chromosomal sequence of *B. mongoliense* BMONG18, confirming the qPCR observations regarding colonization and persistence of this specific strain in the human gut.

Abundance of *B. mongoliense* BMONG18 at T7 and T10 was also correlated with 16S rRNA gene profiling data obtained for the 10 enrolled individuals of the Milk group at T0 (Supplementary Table 8). This analysis revealed that relative abundance of *Lachnospiraceae* NK4B4 group and *Subdoligranulum* positively correlate with the abundance of *B. mongoliense* BMONG18 (Supplementary Table 9). Notably, these taxa have been observed to be transmitted across the Parmesan cheese's production chain (Fig. 2) and previous studies reported that they are common colonizers of mammalian milk as well as the human gut during both infancy and adulthood[36–40]. This positive covariance of *Lachnospiraceae* NK4B4 group and *Subdoligranulum* with *B. mongoliense* hints at cooperative behavior, perhaps through cross-feeding or another metabolic inter-dependency, perhaps supporting their efficient transmission across the Parmesan cheese production chain and their co-colonization of the human gut.

In order to verify if aged Parmesan cheese can also vector *B. mongoliense* colonization of the human gut, we collected a sample of 12-month-old Parmesan cheese produced in the P1 cheese-making site were *B. mongoliense* BMONG18 was originally isolated. Aliquots of one-year Parmesan cheese produced in the P1 cheese plant were subjected to culturomics experiments, involving strain isolation attempts, using bifidobacterial selective media. Notably, these culturomics trials lead to the isolation of viable colonies belonging to the *B. mongoliense* species. The genome sequences of these *B. mongoliense* isolates displayed a 100% nucleotide identity with *B. mongoliense* BMONG18, further supporting the notion that members of the fresh Parmesan cheese microbiota can persist through ripening.

Our results support the notion that *B. mongoliense* BMONG18 has adapted to permit its transmission across the Parmesan cheese production chain, its stability and viability in ripened cheese and its ability to colonize and persist, at least temporarily, in the human gut. For this reason, while *B. mongoliense* was exploited in this study as a test case for the investigation of

bacterial transmission across the Parmesan cheese production chain to humans, further investigations of the biology of this taxon will be pivotal to extend our limited knowledge regarding its functional role in the human gut environment.

**Conclusions**. Parmesan cheese microbiota has been extensively studied to understand its role in cheese making, including ripening and the development of its organoleptic features[41–43]. Nevertheless, the origin and ecology of bacteria found in Parmesan cheese, as well as their possible engraftment in the human gut microbiota has never been investigated in detail. Here, we applied a multi-omics approach based on 16S rRNA gene and bifidobacterial ITS profiling, as well as metagenome-based genome reconstruction combined with strain-specific primer design and culturomics efforts to demonstrate bacterial transmission linking dairy cattle to the human consumer. We observed that bacteria harbored by the bovine gut microbiota or by the housing environment modulate the microbiota of bovine milk and, consequently, corresponding cheese. Moreover, an in vivo pilot study confirmed the hypothesis that bacteria harbored by Parmesan cheese may colonize and persist in the gut of those individuals who consume this cheese on a daily basis. Our study also demonstrates that daily consumption of cow's milk may support the engraftment of bacteria transmitted across the Parmesan cheese production chain to the human gut and suggests the existence of putative co-operative behavior between certain food-vectored bacteria. Altogether, these results highlight that a thorough understanding of the mechanisms responsible for the origin of the cheese microbiota will be pivotal to appreciate its impact on the complex ecological relations between bacterial populations harbored by the gut of cheese consumers, with possible implications on human health.

## Methods

**Ethical statement**. All experimental procedures and protocols involving animals were approved by the Veterinarian Animal Care and Use Committee of Parma University, and conducted in accordance with the European Community Council Directives dated 22 September 2010 (2010/63/UE). Human participants gave their informed written consent before enrollment. All investigations were carried out following the principles of the Declaration of Helsinki.

**Sample collection**. For the purpose of this study a total of 50 animals from five different husbandries located in Emilia Romagna region in Italy were investigated (Supplementary Table 1). Cows were fed only with hay as set out in the Production Guidelines, with free or fixed housing (Supplementary Table 1). For each cow, fecal samples were collected immediately after defecation, while corresponding MILwere taken directly by hand during evening milking, after the teat-ends were cleaned and disinfected. Moreover, 10 environmental samples were recovered from litters, while three fresh Parmesan Cheese samples were collected by trimming the fresh rind of the Parmesan cheese shapes produced with the sampled milks, for each husbandry. All samples were kept on ice, shipped under sub-zero conditions to the laboratory and stored at −80 °C until further processing.

**Bacterial DNA extraction and 16S rRNA gene sequencing**. Stool and environmental samples were subjected to DNA extraction using the QIAamp DNA Stool Mini kit following the manufacturer's instructions (Qiagen). In contrast, DNA extraction of both milk and PC samples were performed using the DNeasy Mastitis Mini Kit (Qiagen). Partial 16S rRNA gene sequences were amplified from extracted DNA using primer pair Probio_Uni (5′-CCTACGGGRSGCAGCAG-3′) and Probio_Rev (5′-ATTACCGCGGCTGCT-3′), targeting the V3 region of the 16S rRNA gene sequence[44]. Illumina adapter overhang nucleotide sequences were added to the partial 16S rRNA gene-specific amplicons, which were further processed employing the 16S Metagenomic Sequencing Library Preparation Protocol (Part #15044223 Rev. B–Illumina). Amplifications were carried out using a Verity Thermocycler (Applied Biosystems). The integrity of the PCR amplicons was analyzed by electrophoresis on a 2200 TapeStation Instrument (Agilent Technologies, USA). DNA products obtained following PCR-mediated amplification of the 16S rRNA gene sequences were purified by a magnetic purification step involving the Agencourt AMPure XP DNA purification beads (Beckman Coulter Genomics GmbH, Bernried, Germany) in order to remove primer dimers. DNA concentration of the amplified sequence library was determined by a fluorimetric Qubit quantification system (Life Technologies, USA). Amplicons were diluted to a

concentration of 4 nM, and 5 μL quantities of each diluted DNA amplicon sample were mixed to prepare the pooled final library. Sequencing was performed using an Illumina MiSeq sequencer with MiSeq Reagent Kit v3 chemicals. Following sequencing, the .fastq files were processed using a custom script based on the QIIME software suite[45]. Paired-end read pairs were assembled to reconstruct the complete Probio_Uni/Probio_Rev amplicons. Quality control retained sequences with a length between 140 and 400 bp and mean sequence quality score >20 while sequences with homopolymers > 7 bp and mismatched primers were omitted. In order to calculate downstream diversity measures (alpha and beta diversity indices, Unifrac analysis), 16S rRNA operational taxonomic units (OTUs) were defined at 100% sequence homology using DADA2[46]; OTUs not encompassing at least two sequences of the same sample were removed. Notably, this approach allows highly distinctive taxonomic classification at single nucleotide accuracy[46]. All reads were classified to the lowest possible taxonomic rank using QIIME2[45,47] and a reference dataset from the SILVA database[48]. Biodiversity within a given sample (alpha-diversity) was calculated with Chao1 and Shannon indexes. Similarities between samples (beta-diversity) were calculated by unweighted uniFrac[49]. The range of similarities is calculated between values 0 and 1. PCoA representations of beta-diversity were performed using QIIME2[45,47].

**Bacterial DNA extraction and bifidobacterial ITS sequencing**. Stool and environmental samples were subjected to DNA extraction using the QIAamp DNA Stool Mini kit following the manufacturer's instructions (Qiagen). In contrast, DNA extraction of both milk and Parmesan cheese samples were performed using the DNeasy Mastitis Mini Kit (Qiagen). Partial ITS sequences were amplified from extracted DNA using the primer pair Probio-bif_Uni (CTKTTGGGYYCCCKG RYYG) and Probio-bif_Rev (CGCGTCCACTMTCCAGTTCTC), which targets the spacer region between the 16S rRNA and the 23S rRNA genes within the ribosomal RNA (rRNA) locus[24]. Illumina adapter overhang nucleotide sequences were added to the partial ITS amplicons, which were further processed employing the 16S Metagenomic Sequencing Library Preparation Protocol (Part #15044223 Rev. B – Illumina). PCR amplifications as well as library preparation was performed as described above for the 16S rRNA microbial profiling analyses. Following sequencing, the .fastq files were processed using a custom script based on the QIIME software suite[45]. Paired-end read pairs were assembled to reconstruct the complete Probio-bif_Uni/Probio-bif_Rev amplicons. Quality control retained sequences with a length between 100 and 400 bp and mean sequence quality score of >20, while sequences with homopolymers > 7 bp in length and mismatched primers were removed. In order to calculate downstream diversity measures (alpha and beta diversity indices, Unifrac analysis), ITS operational taxonomic units (OTUs) were defined at 100% sequence homology using uclust[50]. All reads were classified to the lowest possible taxonomic rank using QIIME2[45,47] and a reference dataset, consisting of an updated version of the bifidobacterial ITS database[24].

**Shotgun metagenomics**. DNA was fragmented to 550–650 bp using a BioRuptor machine (Diagenode, Belgium). Samples were prepared following the TruSeq Nano DNA Sample Preparation Guide (Part#15041110Rev.D). Sequencing was performed using an Illumina NextSeq 500 sequencer with NextSeq Mid Output v2 Kit chemicals (Illumina Inc., San Diego, CA 92122, USA).

**Analysis of metagenomic datasets**. The generated fastq files were scanned for reads corresponding to *Bos taurus* DNA (which were then removed). Only paired data were further analyzed. The retained reads were analyzed with the METAnnotatorX bioinformatic platform[25] in order to perform metagenomic assembly with SPAdes[51], removal of contigs shorter than 3000 bp, ORF prediction and functional classification based on the NCBI RefSeq database and taxonomic assignment of the assembled contigs. All contigs classified as the same bacterial species were collected in the same GenBank file.

SNP profiling was performed using BWA aln software[52] and the metaSNV bioinformatic tool[53] employing default settings as indicated in the online metaSNV tutorial (http://metasnv.embl.de/tutorial.html).

**Isolation of bifidobacterial and lactobacilli strains**. One gram of sample was mixed with 9 ml of phosphate-buffered saline (PBS), pH = 6.5. Serial dilutions and subsequent plating were performed using the de Man-Rogosa-Sharpe (MRS) agar (Scharlau Chemie, Barcelona, Spain), and for bifidobacteria was also supplemented with 0.05% (wt/col) L-cysteine hydrochloride, 50 μg/ml mupirocin (Delchimica, Italy) and 1% lactose (Merck, Germany). Agar plates were incubated in an anaerobic atmosphere (2.99% $H_2$, 17.01% $CO_2$, and 80% $N_2$) in a chamber (Concept 400; Ruskin) at 37 °C for 48 h. Colonies, chosen based on different size and morphologies were randomly picked in MRS broth and incubated 37 °C in anaerobic chamber for 16 h. DNA was extracted from each isolates through a rapid mechanical cell lysis. Briefly, 1.5 ml of culture were centrifuge at 6000 rpm for 5 min. Cell pellets were resuspended in 1 ml of sterile water and placed in a sterile tubes containing glass beads (Sigma-Aldrich now Merck). The cells were lysed by shaking the mix on Precellys 24 homogenizer (Bertin instruments) for 2 min (Maximum setting) followed by 2 min of static cooling; this step was repeated for three times. The lysate cells were centrifuged at 13,000 rpm for 1 min, and the upper phase that contain the DNA was recovered. The taxonomic identification of

the isolates was performed through the PCR amplification and subsequent sequencing of their 16S rRNA gene using specific primer for bifidobacterial genus, known as BIF-specific (5′-GGTGTGAAAGTCCATCGCT-3′) and 23S_bif (5′-GTCTGC CAAGGCATCCACCA-3′). Each PCR cycling program consisted of an initial denaturation step of 3 min at 94 °C, followed by amplification for 35 cycles as follows: denaturation (30 s at 94 °C), annealing (30 s at 56.5 °C), and extension (1 min at 72 °C). The PCR was completed with a single elongation step (10 min at 72 °C). The resulting amplicons were purified using QIAquick PCR Purification kit (Qiagen).

**Genome sequencing of lactobacilli and bifidobacterial strains.** The genome sequences of *B. mongoliense* BMONG18 as well as *L. delbrueckii* LDELB18P1 and *L. delbrueckii* LDELB18P2 were determined by GenProbio SRL (Parma, Italy) using a MiSeq platform (Illumina, UK). The genome libraries were generated following the TruSeq Nano DNA library Prep protocol (Part No. 15041110 Rev. D). The library samples were then loaded into a Flow Cell V3 600 cycles (Illumina) according to the technical support guide, and generated reads were depleted of adapter sequences, quality-filtered, assembled and subsequently the protein-encoding open reading frames (ORF) were predicted and functionally annotated through the MEGAnnotator pipeline[54]. For each genome pair, a value of ANI was calculated using the program JSpecies, version 1.2.1[55].

**RNA extraction and gene expression analyses.** Aliquots of 30 ml of *B. mongoliense* 2016B cells were grown to an optical density at 600 nm ranging from 0.6 to 0.8. Briefly, cultures were centrifugated at 4000 rpm for 10 min at 4 °C. Cell pellets were resuspended in 1 mL of QIAzoL Lysis Reagent (Qiagen, United Kingdom) and placed in a sterile tube containing glass beads (Sigma-Aldrich now Merck). The cells were lysed by shaking the mix on Precellys 24 homogenizer (Bertin instruments) for 2 min (Maximum setting) followed by 2 min of static cooling; this step was repeated for three times. The lysate cells were centrifuged at 12,000 rpm for 15 min, and the upper phase was recovered. The RNA samples were purified using RNAesy Mini Kit (Qiagen, United Kingdom) following the manufacturer's protocol. The quality of the RNA was checked by Tape station 2200 (Agilent Technologies, USA) analysis. RNA concentration and purity were evaluated by Picodrop microliter spectrophotometer (Picodrop, UK).

**RNAseq analysis performed by NextSeq Illumina.** For RNA sequencing, 2 µg of total RNA was treated to remove ribosomal RNA by the Ribo-Zero Magnetic Kit (Illumina), followed by purification of the rRNA-depleted sample by ethanol precipitation. RNA was processed according to the manufacturer's protocol. The yield of rRNA depletion was checked by a Tape station 2200 (Agilent Technologies). Then, 500 ng of an rRNA-depleted RNA sample was fragmented using a Bioruptor NGS ultrasonicator (Diagenode, USA) followed by size evaluation using a Tape station 2200. A whole transcriptome library was constructed using the TruSeq Stranded RNA LT Kit (Illumina). Samples were loaded into a Flow cell V2 75 cycles (Illumina) as indicated by the technical support guide. The reads were depleted of adapters, quality filtered (with overall quality, quality window, and length filters) and aligned to the *B. mongoliense* 2016B reference genome through BWA[52]. Counts of reads that correspond to ORFs were performed using HTSeq (http://www.huber.embl.de/users/anders/HTSeq/doc/overview.html) and analysis of the RPKM values was performed using the formula RPKM = numReads/(geneLength/1000 * totalNumReads/1,000,000).

**In vivo pilot study.** Twenty enrolled individuals were divided in two groups of 10 individuals, named Milk and No-Milk. Both groups consumed the daily suggested dose of Parmesan cheese[33], i.e., 45 g/day for 7 days. Furthermore, individuals in the Milk group drank 200 ml of milk each day for the duration of the study, while the No-Milk group did not consume milk for the duration of the study. Faecal samples were collected before starting of the Parmesan cheese consumption (T0) and at day 3 (T3) and day 7 (T7) of cheese consumption. Moreover, the enrolled individuals were followed for seven additional days and faecal samples were collected at day 10 (T10) and day 14 (T14) from the start of the pilot study. All samples were kept on ice, shipped to the laboratory and stored at −80 °C until they were processed.

**Quantitative PCR.** qPCR was performed using the strain-specific primers Mongoliense_55_Fw (5′-TCGTCAATATGCTGCCGTTG-3′) and Mongoliense_55_Rv (5′-GGCAACAATTCCGGGAGATC-3′). qPCR was performed using iTaq Universal SYBR Green Supermix (Bio-Rad, Hercules, CA) on a CFX96 system (BioRad, CA, USA). qPCR products were detected with SYBR green fluorescent dye and amplified according to the following protocol: one cycle of 95 °C for 2 min, followed by 42 cycles of 95 °C for 5 s and 58 °C for 30 s. The melting curve was 58 °C to 95 °C with increments of 0.5 °C/s. In each run, negative controls (no DNA) were included. DNA extracts from cultures of the strain *B. mongoliense* BMON18 were used for standard curves that was built using the CFX96 software (Bio-Rad, Hercules, CA). Samples were performed in triplicates.

**Statistical analyses.** Statistical analyses were performed with SPSS software v. 22 (IBM SPSS Statistics for Windows, Version 22.0. Armonk, NY: IBM Corp.). *T*-tests

were performed to compare the Chao1 alpha diversity between Milk and Cow Feces as well as LIT, and between Parmesan Cheese and Cow Feces as well as LIT. Adonis test was performed for beta-diversity data of Cow Feces, Litter, Milk, and Parmesan Cheese samples. In addition, *T*-tests were performed to compare qPCR data obtained from faecal samples of Milk and No-Milk individuals at T3, T7, T10, and T14. Paired *t*-tests were performed between 16S rRNA gene microbial profiling data of samples collected at T0 and T7, as well as T0 and T14 from Milk and No-Milk individuals.

**Reporting summary.** Further information on experimental design is available in the Nature Research Reporting Summary linked to this article.

## Data availability

Raw sequences of 16S rRNA gene profiling and bifidobacterial ITS profiling as well as shotgun metagenomics sequence data that support the findings of this study have been deposited in SRA database with the accession codes SRP155009 and SRP167296. *B. mongoliense* BMON18, *L. delbrueckii* LDELB18P1 and *L. delbrueckii* LDELB18P2 genomes data that support the findings of this study have been deposited in GenBank database with the accession codes QRAJ00000000, SETJ00000000 and SETI00000000. *B. mongoliense* BMON18, *L. delbrueckii* LDELB18P1 and *L. delbrueckii* LDELB18P2 isolates that support the findings of this study are available from the corresponding author upon request. The source data underlying Figs. 1, 2, 3, 4 and 5 as well as Supplementary Figures 1, 2, 3, 4 and 5 are provided as a Source Data file.

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

## Acknowledgements

This work was primarily funded by the EU Joint Programming Initiative – A Healthy Diet for a Healthy Life (JPI HDHL, http://www.healthydietforhealthylife.eu/) to DvS (in conjunction with Science Foundation Ireland [SFI], Grant number 15/JP-HDHL/3280) and to M.V. (in conjunction with MIUR, Italy). The study is supported by Fondazione Cariparma, under TeachInParma Project (DV). We furthermore thank GenProbio srl for financial support of the Laboratory of Probiogenomics. This research benefited from the HPC (High Performance Computing) facility of the University of Parma, Italy. We thank Caseificio La Lovetta for its support in collecting animal, environmental and Parmesan cheese samples used in this study.

## Author contributions

C.M. performed bioinformatics and statistical analyses and wrote the manuscript. S.N., G.A., R.A., G.L. and M.M. collected the samples and performed 16S profiling, ITS profiling and shotgun metagenomics sequencing. S.D. and A.V. performed RNASeq analyses. L.M. and G.A.L. performed bioinformaics analyses. S.B., M.C.O. and D.V.S. participated in data analysis and wrote the manuscript. M.V. and F.T. wrote the manuscript and designed the study.

## Additional information

**Competing interests:** The authors declare no competing interests.

