## [Peer Review File · Nature Communications]

Reviewers' comments:

Reviewer #1 (Remarks to the Author):

In this manuscript, authors aimed to study bacterial composition from the Parmesan food chain in five different cheese making sites. Samples were collected from cow feces and their corresponding milk. Samples from litters and fresh cheese were also collected. Authors claimed that microbial transmission events at strain accuracy levels from bovine to human occurred and they identified a core microbiota that was shown to be vertically transmitted. Authors hypothesized that specific genes associated with milk carbohydrate metabolism allowed this transmission and the persistence in the gut of human cheese consumers.

This paper addresses an interesting topic (strain tracking across a food chain until consumers gut) using multi-omics approach but there are major limitations that need to be addressed.

1) Strain level: After assembling metagenomics reads from 8 samples taken different sites and from different cows. Authors design primers to track strains. However, it is unlikely that cow feces from different sites from different city shared the exact same strains by species. This question about the primer specificity. I am wondering why Authors did not apply metagenomics analysis on the same site with paired cow-milk samples. In addition, they are now plenty of software which can help to track strain from metagenomics data (MIDAS, StrainPhlan, metaSNV, DESMAN, ...). Authors would need to use bioinformatics tools to track strains at SNP levels on samples derived from the same food chain.

2) Microbial ecology analysis and natural variation: Main and supplementary figures showed pooled data by sites and by types of samples. It is difficult to grasp the natural variation between sites and between samples. I would suggest authors to show figures with all variations and to assess effect of sites and types of samples on global bacterial composition variations.

3) Core microbiota: As authors wrote L-248 Lactobacillus and Streptococcus strains are added during the process and were not coming from the cow itself. Finally, the "core" microbiota (at genus level) is made with Bifidobacterium and Bacteroides genera only. Consequently, the statement in the introduction that "core microbiota [...] was shown to be vertically transmitted" is quite overselled.

4) Title: "The human gut microbiota is shaped by the engraftment of bovine bacteria vectored by milk products". The word "shaped" supposed that human gut microbiota is modified by the strains contained in the cheese during the clinical study. At this point, the study showed only that a cheese species tracked by qPCR persisted in human gut microbiota. Authors need to provide a gut microbiota comparative analysis between the two study groups.

I have also different detailed comments:

1) Method section: globally statistical analysis is not very described. For instance, P-value are reported in Figure 5 but not the statistical test.

2) Rarefaction curve: does sample were pooled? If yes, pools needed to be split to check whether sequencing depth is enough per sample to reach the plateau.

3) Shannon index: I did not see any results based on Shannon index. Could you provide this as stated in method section?

4) L201: the lower complexity of MIL and PC samples need to be evaluated statistically. A test is missing here.

5) L210: Beta-diversity "partial overlap" does these sample coming from the same sites? Please clarify your figure and statement here.

6) L:212: "limited effect" a statistical test is missing here. Globally, I would suggest authors to add a within/between class analysis to quantify the effect of several factors on microbiota variation (Adonis or Amova method can be used for instance).

- 7) L220: "similar profile" again this statement need to be tested.
- 8) Figure 2 failed to show us heterogeneity between sample type and sites because sample were pooled. In addition, authors did not show how sequencing depth per sample could affect this figure.
- 9) L81 "transmission" I think that authors should globally tone down this statement as they only have observed "shared species" using 16S/ITS and not "strain transmission"
- 10) L-333: as the bioinformatic tool manuscript is only submitted at this step, Authors should describe at least in more detail how they reconstruct strain genome and what quality check they perform to insure that genome is not chimeric (e.g. contigs made with reads from different strains).
- 11) L338: it is not clear if the assembly were done per sample or using a cross assembly step. How authors can delineate strain within species? Authors should clarify this.
- 12) L340: it is not clear how well those primers were specific. "absence of specificity" do you mean "absence of hit" instead?
- 13) Figure 4 is not clear. Does it correspond to one site only or a pool? knowing that primers were designed using samples from different site it is difficult to follow the logic here.
- 14) L430 "culturomics" what do you mean by culturomics, this is not defined in the method? I guess it is strain isolation with DNA sequencing but from the method section is not very clear.
- 15) Figure 5 is not clear. I understood from the method section that only 20 individuals were included into the clinical trial. What about the other 20 non- cheese consumers? This figure could confuse the reader. Bottom panel: there is no indication of subject variation per time point. Authors should put at least a standard deviation.

Reviewer #2 (Remarks to the Author):

Authors present an interesting study characterizing microbiomes and bacterial species from samples across cheese production chain. They use impressive assortment of omics techniques to characterize the microbial community profiles in bovine feces, litters, cow's milk and cheeses manufactured using the milk, which makes the study unique. Authors also claim microbial transmission events on strain level accuracy but part of the data supporting these claims needs further clarification. Authors also conducted a small human intervention study and showed that *B. mongoliense* strains present in cheese can transiently colonize the human gut during cheese consumption.

Major comments:

I found the manuscript relatively long and somewhat repetitive. For example, sections describing 16S rRNA sequencing and ITS sequencing results are very similar by structure but are presented sequentially (this is reflected by similarity of Figs. 2 and 3). Consider combining these sections to make the manuscript more compact. Alternatively consider moving some parts to supplement.

Authors use an outdated computational method, *uclust* (from 2010), to analyze 16S sequencing data. Independent comparisons operating with mock community data have shown that *uclust* is prone to false positive OTUs (see e.g. PMID: 27822515). Authors should process the 16s amplicon data using an up-to-date method such as *DADA2* (PMID: 27214047) or *deblur* (PMID: 28289731).

I was surprised to see high abundances of *B. longum infantis* in milk samples. Is it known that cow's milk contains this (human) infant gut commensal? Do you think these strains can readily engraft in the human gut (and utilize human milk oligosaccharides) or are they cow-specific strains?

Designing primers based on metagenomic assemblies to show strain level similarity poses circular reasoning. Given that the primers were present in the metagenomes, it's not surprising that PCR amplification results in the detection of these species. However, that does not rule out genomic

differences between these genomes in loci outside of the primer sequence. Please, give details on why do you think your approach results in detection of only a particular strain of the given species, not any others. I would argue that it is almost impossible to guarantee that a short primer sequence is specific at strain level, and will not pick up any other strains of the given species (i.e. sensitivity is not a problem but specificity).

Searching the provided accession codes (SRP155009, QRAJ00000000) in NCBI SRA gave no results. Data needs to be made available prior to publication.

Minor comments

Figure 1A, top right panel (PC1 vs. PC2) is enough for making the conclusions authors present in the text. Presenting 3D graph on 2D paper is not feasible and does not provide any added value here.

Figure 1B is unreadable due to large number of (almost) similar colors. Possible to show on higher taxonomic level, only show taxa of interest, or some other way to simplify?

Lines 248-251: This sentence is confusing, please re-write.

Lines 237-240: This is not that surprising to me. Direct-contact bacterial transmission happens all the time. For example, gut bacteria are commonly exchanged between humans sharing the same living environments, why not cows?

Line 333: typo: where -> were

Figure 4: Color legend is missing. (red: presence, black: absence?)

Reviewer #3 (Remarks to the Author):

The authors present a well written, well researched manuscript. I have the following comments.

- when referring to the figures throughout please use a combination of the figure number and letter. i.e. fig 1a rather than 1.
- Does *B. monogoliense* offer any advantage to host health. Perhaps the authors could comment on this.
- in the methods can the authors make a clearer distinction on which methods were ITS and which 16S
- for the shotgun analysis it seems the authors only used this data for primer design; I feel this is a missed opportunity and perhaps a strain level analysis could have been looked at for example with Strainphlan
- for the intervention in the group that dont eat milk do they also refrain from other dairy products? could this be a confounder?
- Also for the intervention study it would have been nice to see compositional analysis of the gut microbiome to see if the rest of the microbiome was altered.
- While the authors were looking at ITS it would have been beneficial to look at fungal populations.
- Were the sequence reads uploaded to a data repository?
- line 257 what was rationale for clustering at 99% and not standard 97(this reviewing understands it was for species but perhaps make this clear in the text.
- line 332 misspelling oth through

Reviewer 1:

Major comments:

1) Strain level: After assembling metagenomics reads from 8 samples taken different sites and from different cows. Authors design primers to track strains. However, it is unlikely that cow feces from different sites from different city shared the exact same strains by species. This question about the primer specificity. I am wondering why Authors did not apply metagenomics analysis on the same site with paired cow-milk samples. In addition, they are now plenty of software which can help to track strain from metagenomics data (MIDAS, StrainPhlan, metaSNV, DESMAN, ...). Authors would need to use bioinformatics tools to track strains at SNP levels on samples derived from the same food chain.

REPLY: We understand the criticism of the reviewer. In fact, primers obtained from assembled metagenomic data of CF, LIT, MIL and PC samples were used to try to accurately trace the occurrence of the corresponding bacterial strains across samples of the same cheesemaking site and not from samples collected from other plants. We better detailed this in the text (Page 17, lines 378-384). Furthermore, we exploited the software metaSNV to validate results obtained from strain-specific primers. In detail, we exploited the shotgun datasets obtained for LIT and PC samples of P1 as well as P2 cheesemaking sites (Table S4) to validate the distribution of strains assembled from these datasets. Notably, the generated data confirmed the results obtained from DNA amplification with strain-specific primers (Page 17-18, lines 389-406).

2) Microbial ecology analysis and natural variation: Main and supplementary figures showed pooled data by sites and by types of samples. It is difficult to grasp the natural variation between sites and between samples. I would suggest authors to show figures with all variations and to assess effect of sites and types of samples on global bacterial composition variations.

REPLY: As suggested by the reviewer, Fig. 1, Fig. S1 and Fig. S2 have been re-typed and reformulated. Moreover, Fig. S3 and Fig. S4 have been added to show data collected from all assessed samples in order to report the variation between sites and samples. We also clarified in the text that the complete profiles at genus level are available in Supplementary Excel file 1 (Page 11, lines 222-223; Fig. S1; Fig. S2; Fig. S3; Fig. S4). Please also note that the analysis of 16S rRNA gene-based microbial profiling data has been completely redone based on the suggestions made by reviewer 2 (see comment 2).

3) Core microbiota: As authors wrote L-248 *Lactobacillus* and *Streptococcus* strains are added during the process and were not coming from the cow itself. Finally, the “core” microbiota (at genus level) is made with *Bifidobacterium* and *Bacteroides* genera only. Consequently, the statement in the introduction that “core microbiota [...] was shown to be vertically transmitted” is quite overselled.

REPLY: As also highlighted by reviewer 2, our statement reported in the text was not clear. For this reason, the text has been re-written to clarify that *Lactobacillus* and *Streptococcus* dominate in the average PC microbiota since they are also added during whey inclusion in addition to *Lactobacillus* and *Streptococcus* strains transmitted from CF, LIT and MIL samples. Nevertheless,

all the genera constituting the core microbiota of the Parmesan cheese production chain are transmitted from CF, LIT and MIL samples to Parmesan cheese. This was confirmed by PCR (Figure 4) targeting sequences of *Lactobacillus* and *Streptococcus* genomes that had been assembled from shotgun metagenomics data of PC samples and that had also been detected in CF, LIT and MIL matrices collected in the same cheese production site (Page 13, lines 279-290). Please note that the analysis of 16S rRNA gene microbial profiling data has been completely redone based on the suggestions made by reviewer 2 (see comment 2).

4) Title: "The human gut microbiota is shaped by the engraftment of bovine bacteria vectored by milk products". The word "shaped" supposed that human gut microbiota is modified by the strains contained in the cheese during the clinical study. At this point, the study showed only that a cheese species tracked by qPCR persisted in human gut microbiota. Authors need to provide a gut microbiota comparative analysis between the two study groups.

REPLY: We agree with the reviewer that additional data is needed to verify that the human gut microbiota is modulated by the strains contained in the Parmesan cheese during the *in vivo* pilot study. For this reason, 16S rRNA gene-based microbial profiling analyses were performed on microbial DNA extracted from all faecal samples collected at T0, T7 and T14 from individuals enrolled in the pilot study, thus extending our previous 16S rRNA gene profiling analyses that had been performed for the 10 enrolled individuals of the Milk group at T0. Results are discussed in the text (Supplementary Excel File 4, Figure S5, Supplementary Figure legends, Pages 20-21, lines 461-474).

Minor comments:

1) Method section: globally statistical analysis is not very described. For instance, P-value are reported in Figure 5 but not the statistical test.

REPLY: As suggested by the reviewer, we have now reported the details regarding the statistical analyses performed in the manuscript (Pages 9-10, lines 197-204).

2) Rarefaction curve: does sample were pooled? If yes, pools needed to be split to check whether sequencing depth is enough per sample to reach the plateau.

REPLY: We followed the reviewer's suggestion and have provided rarefaction curves for each sample included in this study that was subjected to 16S rRNA gene microbial profiling analyses (Fig. S1 and Fig. S2).

3) Shannon index: I did not see any results based on Shannon index. Could you provide this as stated in method section?

REPLY: We have now reported results regarding alpha-diversity evaluated using the Shannon index (Page 11, lines 226; Fig. S1; Fig. S2).

4) L201: the lower complexity of MIL and PC samples need to be evaluated statistically. A test is missing here.

REPLY: As indicated by the reviewer, we have now included data regarding the t-test performed to evaluate if lower complexity of MIL and PC samples, as compared to CF and LIT samples, is statistically significant (Page 11, lines 226-227). Furthermore, we have reported details regarding this statistical analysis in the materials and methods section (Pages 9-10, lines 197-204).

5) L210: Beta-diversity “partial overlap” does these sample coming from the same sites? Please clarify your figure and statement here.

REPLY: As requested by the reviewer, we clarified this concept in the text and added Figure S4 showing samples collected from the same production site (Page 12, line 237-238; Figure S4).

6) L:212: “limited effect” a statistical test is missing here. Gloabaly, I would suggest authors to add a within/between class analysis to quantify the effect of several factors on microbiota variation (Adonis or Amova method can be used for instance).

REPLY: We performed an Adonis statistical test, as suggested by the reviewer, and have described the results in the text (Page 12, lines 241-242).

7) L220: “similar profile” again this statement need to be tested.

REPLY: We agree with the reviewer and have rephrased the sentence (Page 23, lines 248-251).

8) Figure 2 failed to show us heterogeneity between sample type and sites because sample were pooled. In addition, authors did not show how sequencing depth per sample could affect this figure.

REPLY: As suggested by the reviewer, we have provided more details regarding single samples and their sequencing depth (Supplementary Excel file 3). Moreover, the impact of sequencing depth on number of OTUs predicted has been discussed in the text (Pages 14, lines 309-312). The extensive amount of data regarding shared OTUs for each analyzed sample was reported in a supplementary Excel file (Supplementary Excel file 3). In addition, we report in Figure 2 the pooled results since bovine samples from the same cheesemaking site live together and shared the same foods, water and litters, thus inducing extensive cross-contamination. For this reason, we focused on total counts of shared OTUs obtained from analysis of all samples of the same matrix and reported results obtained for each production site.

9) L81 “transmission” I think that authors should globally tone down this statement as they only have observed “shared species” using 16S/ITS and not “strain transmission”

REPLY: We rephrased the text as indicated by the reviewer (Page 145 line 3163).

10) L-333: as the bioinformatic tool manuscript is only submitted at this step, Authors should describe at least in more detail how they reconstruct strain genome and what quality check they perform to insure that genome is not chimeric (e.g. contigs made with reads from different strains).

REPLY: We agree with the Reviewer. Now METAnnotatorX has been accepted for publication (Milani et al., 2018, Microbiome), thus, it has been fully referenced in the manuscript.

11) L338: it is not clear if the assembly were done per sample or using a cross assembly step. How authors can delineate strain within species? Authors should clarify this.

REPLY: As indicated by the reviewer, details regarding metagenomic assembly and taxonomic classification of the retrieved contigs are now available in the recently published METAnnotatorX manuscript (Milani et al., 2018, Microbiome), thus this has been referenced in the text. Moreover, we explained in the text that primers were designed on the largest contig obtained for each traced species, therefore representing a genomic region of a specific strain of that species (Page 17, lines 311-373).

12) L340: it is not clear how well those primers were specific. “absence of specificity” do you mean “absence of hit” instead?

REPLY: As requested by the reviewer, we have better detailed that no hits were observed when primers were tested against the NCBI nt database by means of the Primer BLAST web tool (Page 17, lines 376).

13) Figure 4 is not clear. Does it correspond to one site only or a pool? knowing that primers were designed using samples from different site it is difficult to follow the logic here.

REPLY: Following the reviewer’s suggestion, we improved the clarity of Figure 4 by reporting on the right side of the heat map the cheesemaking site where the genomes were assembled and traced using specific primers (Figure 4; Page 31, line 761).

14) L430 “culturomics” what do you mean by culturomics, this is not defined in the method? I guess it is strain isolation with DNA sequencing but from the method section is not very clear.

REPLY: We have detailed the meaning of culturomics efforts in the text, as requested by the reviewer (Page 21, lines 504-505).

15) Figure 5 is not clear. I understood from the method section that only 20 individuals were included into the clinical trial. What about the other 20 non- cheese consumers? This figure could confuse the reader. Bottom panel: there is no indication of subject variation per time point. Authors should put at least a standard deviation.

REPLY: We agree with the reviewer and clarified Figure 5. Furthermore, we added standard deviation to the graph (Figure 5).

Reviewer 2:

Major comments:

1) I found the manuscript relatively long and somewhat repetitive. For example, sections describing 16S rRNA sequencing and ITS sequencing results are very similar by structure but are presented sequentially (this is reflected by similarity of Figs. 2 and 3). Consider combining these sections to make the manuscript more compact. Alternatively consider moving some parts to supplement.

REPLY: As proposed by the reviewer, we merged the sections presenting 16S OTUs as well as ITS data and significantly shortened the text (Pages 14-16, lines 293-353).

2) Authors use an outdated computational method, uclust (from 2010), to analyze 16S sequencing data. Independent comparisons operating with mock community data have shown that uclust is prone to false positive OTUs (see e.g. PMID: 27822515). Authors should process the 16s amplicon data using an up-to-date method such as DADA2 (PMID: 27214047) or deblur (PMID: 28289731).

REPLY: We agree with the reviewer, thus 16S rRNA gene-based microbial profiling data presented in the manuscript was re-analyzed using DADA2/deblur. Therefore, the results described in the text and the figures regarding 16S rRNA gene-based microbial profiling analyses and OTUs have been updated accordingly (Page 6, lines 109-117; Pages 11-15, lines 213-315; Figure 1, Figure S1, Figure S2; Figure S3; Figure S4).

3) I was surprised to see high abundances of *B. longum infantis* in milk samples. Is it known that cow's milk contains this (human) infant gut commensal? Do you think these strains can readily engraft in the human gut (and utilize human milk oligosaccharides) or are they cow-specific strains?

REPLY: As highlighted by the reviewer, *Bifidobacterium longum* subsp. *infantis* has been extensively studied in humans. Nevertheless, ITS bifidobacterial profiling analyses have previously been employed to explore bifidobacterial distribution across the mammalian branch of the tree of life (Milani et al., 2017, ISME J.) and the data collected revealed that bifidobacterial species seem to have evolved to colonize a wide range of mammalian species, putatively resulting in strain adaptation to specific hosts (Milani et al., 2017, ISME J.). This may explain presence of *B. longum* subsp. *infantis* in cow's milk but its ability to colonize and persist in the human gut, while possible, cannot be confirmed with data collected in this study. We added a discussion on this topic in the manuscript (Page 15, lines 324-326).

4) Designing primers based on metagenomic assemblies to show strain level similarity poses circular reasoning. Given that the primers were present in the metagenomes, it's not surprising that PCR amplification results in the detection of these species. However, that does not rule out genomic differences between these genomes in loci outside of the primer sequence. Please, give details on why do you think your approach results in detection of only a particular strain of the given species, not any others. I would argue that it is almost impossible to guarantee that a short primer sequence is specific at strain level, and will not pick up any other strains of the given species (i.e. sensitivity is not a problem but specificity).

REPLY: We understand the criticism of the reviewer, since primer tracing offers high sensitivity but may not provide high specificity against very closely related strains (in case the amplified region represents a conserved sequence between such closely related strains), thus the limitations of this approach has been outlined in the text (Page 17, lines 387-389). For this reason, and in accordance with the suggestion of reviewer 1, we confirmed our results using strain-specific primers through analysis of the currently available shotgun metagenomics data by metaSNV software (Page 17, lines 389-403). Moreover, we underlined in the text that strain-specific transmission of bacteria across the Parmesan cheese production chain was also verified through strain isolation results (Page 17, lines 404-406).

Searching the provided accession codes (SRP155009, QRAJ00000000) in NCBI SRA gave no results. Data needs to be made available prior to publication.

REPLY: As requested by the reviewer, we have released of our submitted data deposited at the SRA database.

Minor comments:

1) Figure 1A, top right panel (PC1 vs. PC2) is enough for making the conclusions authors present in the text. Presenting 3D graph on 2D paper is not feasible and does not provide any added value here.

REPLY: We agree with the reviewer and we removed the unnecessary 2D representations from Figure 1 panel a. However, we kept the 3D representation because we are convinced it allows the readers to have a complete overview of the PCoA results (Figure 1).

2) Figure 1B is unreadable due to large number of (almost) similar colors. Possible to show on higher taxonomic level, only show taxa of interest, or some other way to simplify?

REPLY: As requested by the reviewer, since Figure 1 panel b already reports taxa with abundance of >5 %, we added a sheet in Supplementary Excel file 1 reporting data showed in Figure 1. This was also indicated in the text and figure legend (Page 30, lines 741).

3) Lines 248-251: This sentence is confusing, please re-write.

REPLY: As also underlined by reviewer 1, the statement reported in the manuscript was not clear. For this reason, it has been completely rephrased (Pages 13-14, lines 283-290).

4) Lines 237-240: This is not that surprising to me. Direct-contact bacterial transmission happens all the time. For example, gut bacteria are commonly exchanged between humans sharing the same living environments, why not cows?

REPLY: As proposed by the reviewer, the sentence has been rephrased to reflect the fact that bacterial transmission between cohabiting individuals has already been reported (Page 13, lines 265-270).

5) Line 333: typo: where -> were

REPLY: We corrected the typo (Page 17, line 365).

6) Figure 4: Color legend is missing. (red: presence, black: absence?)

REPLY: As indicated by the reviewer, we included a color legend in Figure 4.

Reviewer 3:

Major comments:

1) When referring to the figures throughout please use a combination of the figure number and letter. i.e. fig 1a rather than 1.

REPLY: As suggested by the reviewer, figures have been referenced with the appropriate panel throughout all the manuscript.

2) Does *B. mongoliense* offer any advantage to host health. Perhaps the authors could comment on this.

REPLY: We agree with the reviewer that, despite the fact that *B. mongoliense* was chosen as a test case to study bacterial transmission across the Parmesan cheese production chain, a discussion regarding the biological role of *B. mongoliense* in the human gut may be of interest to the reader. However, current knowledge regarding this taxon is limited to a small number of descriptive works, and for this reason further study of *B. mongoliense* is required to gain insights into its biology and its potential cross-talk with human host. A discussion has been added to the manuscript (Page 22, lines 512-516).

3) In the methods can the authors make a clearer distinction on which methods were ITS and which 16S.

REPLY: As indicated by the reviewer, we separated the 16S rRNA gene and ITS profiling methods (Pages 5-7, lines 87-136).

4) For the shotgun analysis it seems the authors only used this data for primer design; I feel this is a missed opportunity and perhaps a strain level analysis could have been looked at for example with Strainphlan.

REPLY: We agree with the reviewer. As also proposed by reviewer 1 (see comment 1), we exploited shotgun metagenomics data to verify strain transmission. In detail, MetaSNV software was employed to perform strain tracing based on available shotgun metagenomics data and genomes reconstructed from metagenomic assembly. Data collected confirmed previous observations obtained from strain-specific primer tracing (Page 17-18, lines 389-406).

5) For the intervention in the group that dont eat milk do they also refrain from other dairy products? could this be a confounder?

REPLY: In accordance to the reviewer, we enrolled in the No Milk group only individuals that did not eat any dairy product during the study. This has now been more clearly outlined in the text (Pages 20, lines 459).

6) Also for the intervention study it would have been nice to see compositional analysis of the gut microbiome to see if the rest of the microbiome was altered.

REPLY: As proposed by the reviewer, and also in accordance to reviewer 1, the 16S rRNA gene-based microbial profiling was performed on DNA extracted from all faecal samples collected at T0, T7 and T14 from individuals enrolled in the pilot study. The achieved results were discussed

in the text (Supplementary Excel File 4, Figure S5, Supplementary Figure legends, Pages 20-21, lines 461-474).

7) While the authors were looking at ITS it would have been beneficial to look at fungal populations.

REPLY: We agree with the reviewer that an in-depth evaluation of the fungal community may be of interest for some readers. However, the main aim of this study was to explore transmission of bacteria through the Parmesan cheese production chain by means of a multi-omics approach, and for this reason we didn't pursue fungal ITS profiling. Precise tracing of the fungal population will be considered for a follow-up study.

8) Were the sequence reads uploaded to a data repository?

REPLY: Reads were uploaded to the SRA database, but their public release was originally set at the publication date of the manuscript. We now requested immediate release of these data.

9) Line 257 what was rationale for clustering at 99% and not standard 97(this reviewing understands it was for species but perhaps make this clear in the text.

REPLY: As indicated by reviewer 2, we redid all the 16S rRNA profiling analysis employing Qiime2 and DADA2 clustering software that was specifically developed to obtain OTUs clustering at 100% in order to perform taxonomic classification with single nucleotide accuracy (Callahan et al., 2016, Nat Methods). We have now more clearly explained the rationale behind this choice in the text (Page 6, lines 111-112).

10) Line 332 misspelling oth through.

REPLY: We corrected the typo indicated by the reviewer (Page 16, line 364).

Reviewers' comments:

Reviewer #1 (Remarks to the Author):

Main comments:

I read carefully the second version of the manuscript entitled "The human gut microbiota is shaped by the engraftment of bovine bacteria vectored by milk products". Reading this title, authors needed to show 1) vertical strains transmissions and 2) the alteration of human gut microbiota by the transmitted strain. Demonstrating those statements would be relevant to the field. However, authors failed to show how specific primers are to claim that they were able to track strains from Cow to Cheese (they still did not show any data about SNP profiling for instance). Authors demonstrated that there is NO difference between individual having or not the strain tracked in their gut. So, at this stage, current article claims cannot be accepted.

Detailed comments:

1) Strain level: First, the use of metaSNV is not described in the method section. Second, I did not see any output data from metaSNV in the manuscript like SNP profiling or SNP distance analysis. Third, strain analysis is the keystone method to support claims in this manuscript (and highlighted by the two other reviewers in the previous round). Authors of this study should show more details about how their primers help to track strain. When cross-assembly is used, it is possible than contig originated from different reads from different closely related strain. Thus, as up now, this statement: "the largest contig, representing a chromosomal region of a specific strain," is not convincing. The largest contig do not guarantee that it originates from the same strain. Authors need to work at read levels to untangle if primers were SNP specific. I suggest 1) to work at SNP levels and 2) to report SNP profile for each sample (using metaSNV output for instance) for this targeted contig. As a side note, the software used "METAnnotatorX" do not allow to perform strain analysis and do not guarantee that contig are not chimeric from different strains.

2) Microbial ecology analysis and natural variation: Figure1: authors should choose to show the 2D or the 3D plot not both. Those plots are somehow redundant. Acronyms (CF, MIL, LIT, PC) were described in the main text but should be also described in the figure legend. Figure1b is useless as it is impossible for readers to make the link between color code (numerous genera color coded) and the barplot. Authors should focus on specific category (those written in the main text for instance) or remove it. Regarding alpha-diversity, I would suggest authors to report index using boxplot per group after read rarefaction at the same levels.

3) From the figure 4, it is not clear for instance to see whether the *L. delbrueckii* from stool is the same strain vertically transmitted in cheese. Authors should align SNP profile from read for those sample to see whether they share the same profile. At this stage primers design was not sufficient to show that they really help to track strains in environmental sample. Instead to claims that there is bacterial transmission, I would suggest authors to write that samples from food chain shared the same species (or subspecies) but not strains.

4) Shaping the Human gut microbiota: as demonstrated by authors "no significant differences were identified between time points and groups by means of paired T-test", so the human gut microbiota is NOT shaped by the bovine bacteria. I would suggest authors to change their title.

Reviewer #2 (Remarks to the Author):

Authors have addressed most of my comments in their revisions. However, the newly added analysis of metagenomic data is confusing and does not currently provide any evidence to substantiate authors' claims as far as I understand.

Authors have conducted additional analysis using metaSNV software which is used for calling metagenomic single nucleotide variants (SNVs). In Table S6, it's not clear to me what is "Genome coverage" (varies between 0-451) and how that demonstrates strain level identity for e.g. *C. stationis* where Genome coverage = 0.0022. I suspect that by restricting the analysis to reads with 100% alignment identity the authors are introducing an artificial bias in their results: the regions where strains differ by SNVs are now left unobserved. I would like to see comparison of strain sequence identities, by comparing all shared genomic regions (not only 100% alignment identity) by sequence identity metric after (e.g. Kimura distance) calling SNVs. In order to claim identical strains, show that the genetic distance is significantly lower in comparison to strains that are known to be non-identical.

The paper will benefit either major revision or removal of this section.

Tommi Vatanen

Reviewer #3 (Remarks to the Author):

The authors have addressed all my comments.

Reviewer 1:

Main comments:

1) I read carefully the second version of the manuscript entitled “The human gut microbiota is shaped by the engraftment of bovine bacteria vectored by milk products”. Reading this title, authors needed to show 1) vertical strains transmissions and 2) the alteration of human gut microbiota by the transmitted strain. Demonstrating those statements would be relevant to the field.

However, authors failed to show how specific primers are to claim that they were able to track strains from Cow to Cheese (they still did not show any data about SNP profiling for instance). Authors demonstrated that there is NO difference between individual having or not the strain tracked in their gut. So, at this stage, current article claims cannot be accepted.

REPLY: We understand the criticism of the reviewer about the title. For this reason, and in accordance with the editor, the title was modified as follows “Colonization of the human gut by bovine bacteria present in Parmesan cheese.” (Page 19, lines 434-435; Page 22, lines 525-527). Moreover, in order to validate the horizontal transmission of bacterial strains across the Parmesan cheese production chain, we followed the metaSNV tutorial (<http://metasnv.embl.de/tutorial.html>) in order to perform a SNP profiling and distance analyses that confirmed our previous results (Supplementary Excel file 4; Page 18, lines 396-427). To obtain a comprehensive and exhaustive validation of these results, we also relied on an *in vitro* approach based on strain isolation from cow feces (CF), litters (LIT), milk (MIL), Parmesan cheese (PC) samples collected from cheese making sites P1 and P2, followed by genomic sequencing (Page 11, lines 213-215; Pages 19-21, lines 440-477). Details are reported in the point by point response to major comments reported below.

Major comments:

1) Strain level: First, the use of metaSNV is not described in the method section. Second, I did not see any output data from metaSNV in the manuscript like SNP profiling or SNP distance analysis. Third, strain analysis is the keystone method to support claims in this manuscript (and highlighted by the two other reviewers in the previous round). Authors of this study should show more details about how their primers help to track strain. When cross-assembly is used, it is possible than contig originated from different reads from different closely related strain. Thus, as up now, this statement: “the largest contig, representing a chromosomal region of a specific strain,” is not convincing. The largest contig do not guarantee that it originates from the same strain. Authors need to work at read levels to untangle if primers were SNP specific. I suggest 1) to work at SNP levels and 2) to report SNP profile for each sample (using metaSNV output for instance) for this targeted contig. As a side note, the software used “METAnnotatorX” do not allow to perform strain analysis and do not guarantee that contig are not chimeric from different strains.

REPLY: We followed the suggestions proposed by the reviewer and we performed additional SNP-level analyses using metaSNV using the settings and protocol indicated in the metaSNV tutorial (<http://metasnv.embl.de/tutorial.html>), as now reported in the methods section (Page 8, lines 147-149). In detail, the shotgun metagenomics datasets obtained from litter and Parmesan cheese samples of cheese making sites P1 and P2, i.e. P1_LIT_4, P1_PC_1, P2_LIT_8

and P2_PC_1, were mapped on the 10 contigs (corresponding to 10 different species listed in Figure 4) assembled from these datasets and used for primer design. Subsequently, metaSNV was used to perform SNP profiling and distance analysis. Results have been reported in the new Supplementary Excel file 4. As expected, SNP profiles confirmed that the assembled contigs correspond entirely to the most abundant strain, as indicated by homogeneous profiles and absence of SNPs with high frequency when reads corresponding to datasets used for assembly are mapped. Moreover, mapping of the additional dataset collected from the same cheese production site provided a very similar SNP profile in all 10 cases, thus confirming that litters and Parmesan cheese samples collected in the same cheese making site share an identical bacterial strain. In contrast, mapping of datasets obtained from samples collected in a cheese production site where a different strain was expected indeed resulted in markedly different SNP profiles, as already shown by the absence of a strain-specific amplicon in the PCR approach (as based on strain-specific primers). Observations based on SNP profiles were also confirmed by distance analysis (Supplementary Excel file 4). These results have now been included in the text (Pages 17-18, lines 396-427).

In order to further corroborate our observations regarding strain transmission across the Parmesan cheese production chain by means of an *in vitro* approach, we performed culture-dependent experiments focusing on culturable bacteria, i.e. lactobacilli and bifidobacteria, directed to isolate and subsequently genetically characterize (through genome sequencing) the isolated strains (Page 8, lines 150-167). Through screening of a total of 82 isolates, we were able to successfully isolate the identical strain of *Lactobacillus delbrueckii* (LDELB18P1) and *Bifidobacterium mongoliense* (BMONG18) in samples of cow feces (CF), milk (MIL), litter (LIT) and Parmesan cheese (PC), all associated with cheese making site P1. Moreover, one specific strain of *L. delbrueckii* (LDELB18P2) was isolated from all matrices of cheese making site P2. Genomic sequencing of these 12 isolates and genomic alignments of putatively identical isolates from CF, MIL, LIT and PC matrices resulted in an Average Nucleotide Identity (ANI) value of 100%. Moreover, genomic alignment of *B. mongoliense* isolated from P1 and *L. delbrueckii* isolated from P2 with the contigs used to design primers for tracing of these two strains in the corresponding cheese making sites resulted in an identity level of 100%, thus confirming SNP profiling data. These results have been included in the text (Page 10, lines 213-215; Pages 19-21, lines 440-477).

2) Microbial ecology analysis and natural variation: Figure1: authors should choose to show the 2D or the 3D plot not both. Those plots are somehow redundant. Acronyms (CF, MIL, LIT, PC) were described in the main text but should be also described in the figure legend. Figure1b is useless as it is impossible for readers to make the link between color code (numerous genera color coded) and the barplot. Authors should focus on specific category (those written in the main text for instance) or remove it. Regarding alpha-diversity, I would suggest authors to report index using boxplot per group after read rarefaction at the same levels.

REPLY: We implemented all recommendations made by the reviewer in order to improve Figure 1. Thus, the 3D plot was removed and the acronyms were explained in the figure legend (Figure 1 and page 32, lines 807-812). Moreover, the bar plot was removed from Figure 1 and 16S rRNA

gene microbial profiling data are now referenced in the text as Supplementary Excel file 1. This Excel file reports a tab with the average relative abundance observed for each genus in all the matrices and sampling sites included in this study (Page 12, lines 252, 255, 257-258 and 261). As requested by the reviewer, data regarding alpha-diversity of each profiled matrix has been used to generate box plots that are now reported in panels b and c of Figure 1 (Figure 1, Page 11, lines 233-241 and page 32, lines 807-812).

3) From the figure 4, it is not clear for instance to see whether the *L. delbrueckii* from stool is the same strain vertically transmitted in cheese. Authors should align SNP profile from read for those sample to see whether they share the same profile. At this stage primers design was not sufficient to show that they really help to track strains in environmental sample. Instead to claims that there is bacterial transmission, I would suggest authors to write that samples from food chain shared the same species (or subspecies) but not strains.

REPLY: We agree with the reviewer's comment and we modified the text to underline that PCR results allowed us to trace 14 species (Page 17, line 389-397), and that subsequent metagenomic analysis allowed verification of the same strain based on SNP profiles and distance analysis obtained with metaSNV software (Pages 17-18, lines 396-427). Moreover, additional isolation and sequencing efforts were employed to verify transmission of *L. delbrueckii* and *B. mongoliense* strains across the matrices in two different cheese making sites (Pages 19-21, lines 440-477).

4) Shaping the Human gut microbiota: as demonstrated by authors "no significant differences were identified between time points and groups by means of paired T-test", so the human gut microbiota is NOT shaped by the bovine bacteria. I would suggest authors to change their title.

REPLY: We agree with the reviewer's comment and we have changed the title accordingly. We have also contacted the editor and we modified the title according to a clear indication of the editor as follow: "Colonization of the human gut by bovine bacteria present in Parmesan cheese".

Reviewer 2:

Major comments:

1 Authors have conducted additional analysis using metaSNV software which is used for calling metagenomic single nucleotide variants (SNVs). In Table S6, it's not clear to me what is "Genome coverage" (varies between 0-451) and how that demonstrates strain level identity for e.g. *C. stationis* where Genome coverage = 0.0022. I suspect that by restricting the analysis to reads with 100% alignment identity the authors are introducing an artificially bias in their results: the regions where strains differ by SNVs are now left unobserved. I would like to see comparison of strain sequence

identities, by comparing all shared genomic regions (not only 100% alignment identity) by sequence identity metric after (e.g. Kimura distance) calling SNVs. In order to claim identical strains, show that the genetic distance is significantly lower in comparison to strains that are known to be non-identical.

REPLY: We agree with the criticism of the reviewer, and have, also in accordance with the requests of reviewer 1, performed additional analyses based on SNP profiling following the tutorial available on metaSNV website (<http://metasnv.embl.de/tutorial.html>). SNP profiles and distance matrices have been reported in the new Supplementary Excel file 4. Briefly, the obtained results confirmed that samples collected from the same cheese making site share the same strains of the 10 profiled species, as indicated by comparable SNP profiles and distance analysis. In contrast, samples of different cheese making sites encompass different strains (Pages 18-19, lines 411-427). These findings provide further proof for the hypothesis of specific bacterial transmission across the Parmesan cheese production chain. In order to further validate this observation, we performed isolation and sequencing efforts that indeed demonstrated the presence of identical *L. delbrueckii* and *B. mongoliense* strains across all the samples matrices, i.e. cow's feces (CF), milk (MIL), litter (LIT) and Parmesan cheese (PC) samples, of cheese makings sites P1 and P2 (Pages 19-20, lines 440-469).

We have further modified the text by separating the previous text in two new paragraphs ("Isolation of horizontally transmitted strains" and "Transcriptomics analysis of *B. mongoliense* BMONG18 strain."), so that the final draft of our manuscript in our opinion is now of a much better quality. Therefore, we hope that the revised contributions have addressed all reviewers' comments. Thank you sincerely for consideration of this revised manuscript.

REVIEWERS' COMMENTS:

Reviewer #1 (Remarks to the Author):

I carefully reread the manuscript entitled now "Colonization of the human gut by bovine bacteria present in Parmesan cheese".

I really appreciated the fact that authors took into consideration my previous remarks, particularly put efforts onto the SNPs analysis from metagenomic samples. In addition, the effort of isolating and sequencing several isolates was remarkable.

There is now enough evidence that same strains can be isolated and tracked from cow fecal samples to the cheese samples.

At this point, I have no further major comments. I would like to make some minors suggestions to improve the manuscript:

1) Figure 4 would benefit from being more precise if it indicates how strain tracking was carried out. Indeed, the strain detection in milk samples could only be carried out by PCR. It would be nice to see a color code that would show by which methods the tracking was done : PCR or metagenomics SNP calling (or both methods).

2) L430 : "10 cases" I am not sure what "cases" means? would you mean "species" ?

3) L432: "These findings confirm that the different sampled matrices, i.e. CF, LIT, MIL and PC, of the same cheese making site share the same bacterial strain (Supplementary Excel file 4)." => this is not true for Milk samples in this supplementary file. Milk samples were tested only by PCR.

Reviewer #2 (Remarks to the Author):

I congratulate the authors for their work. All my concerns have been addressed and I think this is a valuable contribution.

Tommi Vatanen, PhD

Reviewer 1:

Comments:

1) Figure 4 would benefit from being more precise if it indicates how strain tracking was carried out. Indeed, the strain detection in milk samples could only be carried out by PCR. It would be nice to see a color code that would show by which methods the tracking was done : PCR or metagenomics SNP calling (or both methods).

REPLY: We agree with the reviewer. Thus, we modified Figure 4 to show in red, purple or blue when strains were traced using PCR, SNP profiling or both methods (Figure 4 and Page 33, lines 856-858).

2) L430 : "10 cases" I am not sure what "cases" means? would you mean "species" ?

REPLY: We changed the text as suggested by the reviewer (Page 12, line 273)

3) L432: "These findings confirm that the different sampled matrices, i.e. CF, LIT, MIL and PC, of the same cheese making site share the same bacterial strain (Supplementary Excel file 4)." => this is not true for Milk samples in this supplementary file. Milk samples were tested only by PCR.

REPLY: As indicated by the reviewer, we corrected the text (Page 12, line 273)